# About the suitability of different numerical methods to reproduce model wind turbine measurements in a wind tunnel with high blockage ratio

Annette Claudia Klein[1], Sirko Bartholomay[2], David Marten[2], Thorsten Lutz[1], George Pechlivanoglou[2], Christian Navid Nayeri[2], Christian Oliver Paschereit[2], and Ewald Krämer[1]

[1]University of Stuttgart, Institute of Aerodynamics and Gas Dynamics, Pfaffenwaldring 21, 70569 Stuttgart, Germany
[2]TU Berlin, Chair of Fluid Dynamics, Müller-Breslau-Straße 8,10623 Berlin, Germany

*Correspondence to:* Annette Claudia Klein (annette.klein@iag.uni-stuttgart.de)

**Abstract.** In the present paper, numerical and experimental investigations of a model wind turbine with a diameter of 3.0m are described. The study has three objectives. The first one is the provision of validation data. The second one is to estimate the influence of the wind tunnel walls by comparing measurements to simulated results with and without wind tunnel walls. The last objective is the comparison and evaluation of methods of high fidelity namely Computational Fluid Dynamics and medium fidelity namely Lifting Line Free Vortex Wake. The experiments were carried out in the large wind tunnel of the TU Berlin where a blockage ratio of $40\%$ occurs. With the Lifting Line Free Vortex Wake code *QBlade*, the turbine was simulated under far field conditions at the TU Berlin. Unsteady Reynolds-averaged Navier-Stokes simulations of the wind turbine, including wind tunnel walls and under far field conditions, were performed at the University of Stuttgart with the Computational Fluid Dynamics code *FLOWer*.

Comparisons between experiment, the Lifting Line Free Vortex Wake code and the Computational Fluid Dynamics code include on-blade velocity and angle of attack. Comparisons of flow fields are drawn between experiment and the Computational Fluid Dynamic code. Bending moments are a compared between the simulations.

A good accordance was achieved for the on-blade velocity and the angle of attack, whereas deviations occur for the flow fields and the bending moments.

## 1 Introduction

In order to improve wind turbines, new strategies and concepts have been developed over the last couple of years. Prior to their application on real wind turbines, they have to be analyzed in detail and the underlying processes have to be completely understood. In many cases, investigations take place on model wind turbines, which is less expensive than building a full size prototype. Moreover, in wind tunnel tests, reproducible inflow conditions can be created.

Bastankhah and Porté-Agel (2015), for example, investigated the interaction between the wake of turbines under yawed conditions. They used particle image velocimetry (*PIV*) for flow physics studies on this complex interaction phenomenon. In subsequent investigations, see Bastankhah and Porté-Agel (2017), they additionally used hot-wire anemometry to analyze the

flow upsteam of the turbine, as well as in the near-wake and far-wake regions. Chamorro and Porté-Agel (2009) used hot-wire anemometry to characterize, amongst others, the distribution of mean velocity and turbulence intensity in the cross section of a wind tunnel at different locations downwind of a wind turbine. Medici and Alfredsson (2006) examined the wake of a model wind turbine under uniform inflow and under the influence of free stream turbulence in terms of 3D effects. For these

investigations, as well as for the investigations of a model wind turbine under yaw misalignment, two-component hot-wires were used to measure the velocity fields.

Even a micro wind farm can be installed in a wind tunnel to investigate the unsteady loading and power output variability, see Bossuyt et al. (2016, 2017). Howland et al. (2016) used the same experimental setup of the micro wind farm to investigate the power output for a variety of yaw configurations.

Moreover, wind tunnel measurements can be used to validate and further develop numerical codes. In the *MEXICO* project (Schepers and Snel (2007)), comprehensive measurements of a three bladed rotor model of $4.5\text{m}$ diameter have been conducted. The experimental data were used, amongst other, to validate numerical methods. Bechmann et al. (2011), for instance, used the *PIV* data, together with the pressure distribution, to validate their Computational Fluid Dynamics(*CFD*) simulations. Blind tests, for example of unsteady aerodynamics experiment as done in the NASA-Ames wind tunnel (Simms et al., 2001),

can be used to improve the development of wind turbine aerodynamics codes and the provided data can also be used for their validation.

If the model wind turbine is investigated in a closed test section, the wind tunnel walls can influence the results. The extend of this influence depends on the blockage ratio, which is defined as the rotor swept area divided by the wind tunnel cross section. Schreck et al. (2007), as well as Hirai et al. (2008), investigated model wind turbines in wind tunnels with a blockage ratio of

approximately $10\%$ and made no blockage correction. Chen and Liou (2011) quantitatively investigated the effects of tunnel blockage on the power coefficient of a horizontal axis wind turbine in a wind tunnel through experiments. They confirmed the results of Schreck et al. (2007) and Hirai et al. (2008), as they found, that the blockage correction is less than $5\%$ for a blockage ratio of $10\%$. Schümann et al. (2013), who experimentally investigated the wakes of wind turbines in a wind tunnel, also showed that for a blockage ratio smaller than $10\%$, no blockage effect should be experienced and the wind tunnel walls

can be neglected. Sarlak et al. (2016) performed Large Eddy Simulations (*LES*) in order to investigate the blockage effects on the wake and power characteristics of a horizontal-axis wind turbine. Thereby, the turbine was modelled with the actuator line technique. They found, that for the operation of the wind turbine close or above the optimal tip speed ratio, even blockage ratios which are larger than $5\%$ will have a substantial impact on the turbine performance.

Fischer et al. (2018) performed unsteady Reynolds-averaged Navier-Stokes (*URANS*) simulations of a model wind turbine in

a cylindrically shaped wind tunnel. To save computational time, the rotational symmetry of the turbine was exploit and only one third of the rotor was simulated. In such a $120°$-model, periodic boundary conditions are used, solely one blade is taken into account and the tower is neglected. In this wind tunnel, the blockage ratio is $> 50\%$. A strong influence of the wind tunnel walls was experienced leading to a more than $60\%$ increase of the driving forces and $25\%$ of the thrust in average. The full model of the same turbine in the real wind tunnel (blockage ratio $40\%$) was simulated by Klein et al. (2018). Thereby, an

increase of $25\%$ in thrust and $50\%$ in power was experienced.

But until now, the performance of a model wind turbine at such a high blockage ratio has not been verified with experimental data.

Thus, the provision of experimental data for the validation of the numerical approaches is one of the three objectives of the present study. The second is the estimation of the influence of the wind tunnel walls. It will be evaluated by comparing *CFD* simulations with and without wind tunnel walls to experimental data. The third deals with the comparison of codes with different degrees of fidelity.

In the present paper, the same model wind turbine and wind tunnel as used by Klein et al. (2018) will be investigated experimentally and numerically. The studied Berlin Research Turbine (*BeRT*), see Pechlivanoglou et al. (2015), was designed and built by TU Berlin and *SMART BLADE GmbH* with a contribution of TU Darmstadt in the aerodynamic blade design. The measurements are conducted in a circuit wind tunnel and the simulations are performed with two methods with different degrees of complexity. A Lifting Line Free Vortex Wake (*LLFVW*) code (*QBlade*) simulates the turbine under free stream conditions. In the numerical setup of the *CFD* code *FLOWer*, the wind tunnel walls and the nozzle are taken into account, but also a case with far field, where the walls are neglected and the boundaries of the setup are far off, is simulated in order to estimate the influence of the wind tunnel walls and to enable a better comparison to the *QBlade* results.

One baseline case and two different yaw-misalignment cases of the turbine are investigated in this study. All simulations are conducted with uniform inflow. At cutting planes upstream and downstream of the turbine, velocities are compared between experiment and *FLOWer*. The on-blade velocities and angles of attack (AoA), as seen by defined blade sections, are compared between experiment, *QBlade* and *FLOWer*. As the determination of the AoA in *CFD* is complex, two different methods are used in *CFD*. Moreover, the bending moments at the blade root are compared between *QBlade* and *FLOWer*.

The numerical and experimental investigation of the turbine is part of the *DFG PAK 780* project (Nayeri et al., 2015), where six partners from five universities work together in the field of wind turbine load control.

## 2  Methodology and setups

In the following, an overview of the characteristics of the setups is given in subsection 2.1. The experimental setup is described in detail in subsection 2.2, followed by the description of the numerical methods and setups of *QBlade* (subsection 2.3) and *FLOWer* (subsection 2.4).

### 2.1  Overview and general characteristics of the setups

As the paper deals with a multitude of cases and setups, the following subsection gives an overview and summarizes the particular characteristics of the setups.

As, according to Schepers (2012), wind turbines are exposed to yaw misalignment from $2\%$ up to $10\%$ of their operating time, these load cases play an important role in wind energy. Therefore, three different cases concerning the inflow direction are taken into account in the present paper. $CaseBASE$ corresponds to the turbine with no yaw misalignment. In $CaseYAW15$,

the turbine is rotated by $-15°$ (clockwise) around the vertical axis of the rotor plane. Usually, a turbine is rotated around the tower. However, as the model wind turbine is placed in a wind tunnel, a rotation around the tower would lead to different clearance distances of the blades to the wall for one revolution. Therefore, the turbine is rotated around the z-axis of the rotor in order to achieve a constant distance between blade tip and wind tunnel walls over a whole revolution. $CaseYAW30$ is rotated by $-30°$. In all simulations uniform inflow is considered. The experimental results have the affix $_{Exp}$, the ones of *QBlade* $_{QBlade}$ and the *FLOWer* results are designated by $_{FLOWer}$. The far field case of *FLOWer* has the addition $_{-FF}$. Table 1 gives an overview of the different cases.

Fig. 1 shows the surfaces of $CaseBASE_{FLOWer}$ and $CaseYAW30_{FLOWer}$. There, the unusual position of the nozzle,

**Table 1.** Overview of the cases.

| Wind tunnel | | | |
| --- | --- | --- | --- |
| Yaw | Experiment | *QBlade* | *FLOWer* |
| $0°$ | $CaseBASE_{Exp}$ | — | $CaseBASE_{FLOWer}$ |
| $-15°$ | $CaseYAW15_{Exp}$ | — | $CaseYAW15_{FLOWer}$ |
| $-30°$ | $CaseYAW30_{Exp}$ | — | $CaseYAW30_{FLOWer}$ |
| Far field | | | |
| Yaw | Experiment | *QBlade* | *FLOWer* |
| $0°$ | — | $CaseBASE_{QBlade}$ | $CaseBASE_{FLOWer-FF}$ |
| $-15°$ | — | $CaseYAW15_{QBlade}$ | — |
| $-30°$ | — | $CaseYAW30_{QBlade}$ | — |

which will be explained in section 2.2.1, and the uncommon yaw movement become obvious.

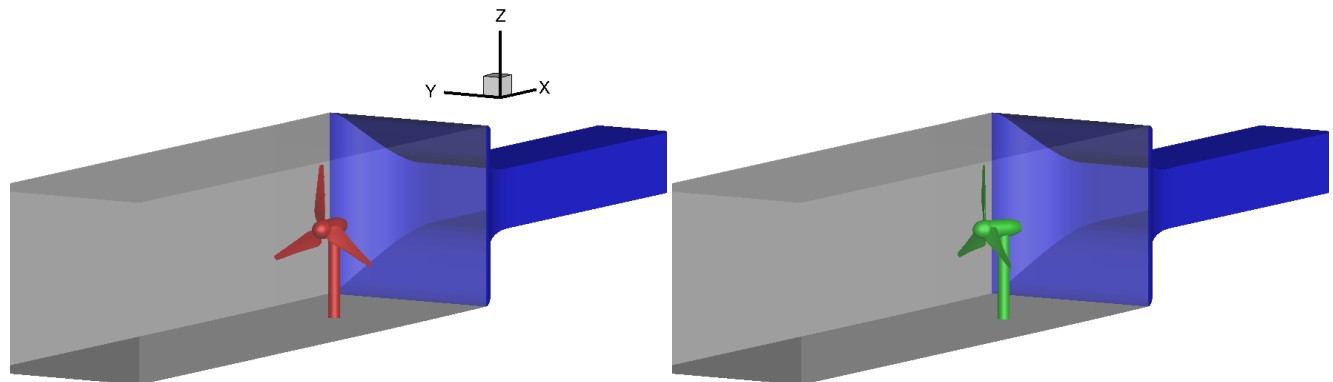

**Figure 1.** Surface for $CaseBASE_{FLOWer}$ (left) and $CaseYAW30_{FLOWer}$ (right).

## 2.2 Experimental setup

The experimental setup consists of the wind tunnel and the model wind turbine, which will be described in the following sections. The blades of the model wind turbine are described in detail in an additional section, as they deliver the data for the comparison with the numerical solutions.

### 2.2.1 Wind tunnel

The experiments are carried out in the large wind tunnel (*GroWiKa*) of the TU Berlin, Fig. 2 (Bartholomay et al., 2017), which is a circuit wind tunnel and is driven by a $450$kW fan. The $2 \times 1.4$m$^2$ cross section of the real test section is too small for the model wind turbine, which has a large diameter to realize the investigation of spanwise locally distributed devices for passive and active flow control in future investigations. Therefore, the real test section was shortened and the $4.2 \times 4.2$m$^2$ settling chamber of the wind tunnel was extended to a total length of $5$m and was then used as measuring section for the model wind turbine. This configuration leads to the unusual fact that the nozzle is positioned downstream of the measuring section. The velocity in the settling chamber used for the present investigations amounts $6.5$ms$^{-1}$ and the turbulence intensity is in average $Ti \leq 1.5\%$ and shows a fairly homogeneous distribution. Three screens are placed upstream of the turbine which aim at increasing the homogeneity in the flow. Additionally, one filtermat is installed at the position of the most upstream screen. Nonetheless, the turbulence intensity is higher in the settling chamber compared to the original test section and the inflow velocity is not perfectly homogeneous. More information about the x-velocity can be found in subsection 4.1 or in Bartholomay et al. (2017). The turbulence in the inflow might lead to a faster recovery of the wake and to higher fluctuations of the loads compared to a case with lower turbulence. As the wind tunnel is short, the influence of the turbulence on the vortex breakdown might be less pronounced than in a far field case or in a longer wind tunnel. Moreover, Medici and Alfredsson (2006) showed, that up to $x/d = 2$, the initial wakes for a case with and without free stream turbulence are quite similar, even with a higher turbulence intensity as in the present setup. However, the blockage ratio by Medici and Alfredsson (2006) was less than $3\%$ and consequently much smaller than in the present case.

### 2.2.2 Berlin Research Turbine (*BeRT*)

The Berlin Research Turbine (*BeRT*), Fig. 3, has a rotor diameter of $3$m with a tower height of $2.1$m. The three blades are exchangeable and equipped with the *Clark-Y* airfoil throughout the complete blade radius from tip to hub. This airfoil has a maximal thickness of $11.8\%$ and was used as it provides attached flow for low Reynolds numbers, as they occur in the blade root region (e.g. $Re_{15\%R} = 170000$). Moreover, it has a good effectiveness of flaps, which will be investigated on the turbine in future experiments and simulations. The twist was chosen so that the local angle of attack stays constant over the span. In order to get a defined transition position for the *CFD* simulations, zig-zag tape has been placed on the blades. The height

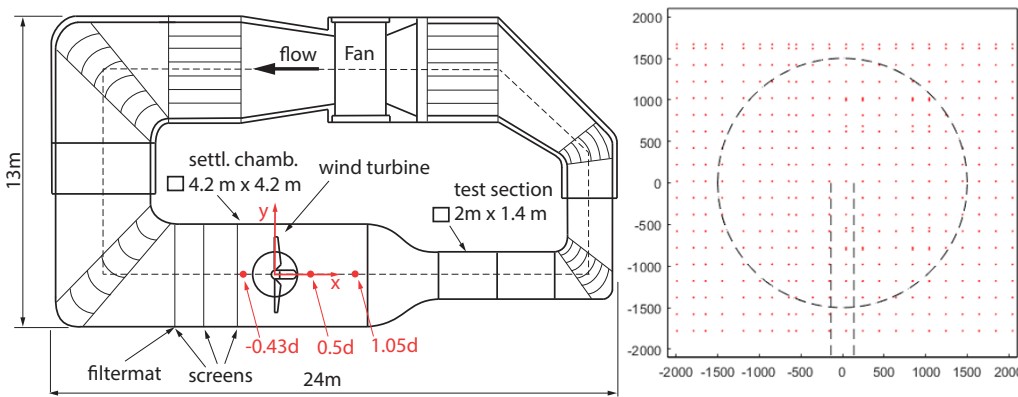

**Figure 2.** Large wind tunnel of the TU Berlin (left) and hot-wire measurement position in each cross-plane (right), (Bartholomay et al., 2017). The dashed lines in the right picture indicate the rotor and the tower.

of the turbulator was estimated experimentally in an additional 2D experiment. It is adapted to the Reynolds number, which varies with the rotor radius, and is consequently staggered. It amounts h=0.75mm inboard up to h=0.21mm outboard on the suction side and h=0.95mm inboard up to h=0.50mm outboard on the pressure side. On the suction side, the leading edge of the tape was positioned at 5% chord, on the pressure side at 10% chord. As the main goal of the turbine is to deliver data for the comparison to simulations and to test and analyze flow control devices and not to compare the overall performance to a turbine in the free field, a realistic scaling was of subordinate interest.

The turbine data is summarized in Table 2 (Bartholomay et al., 2017; Pechlivanoglou et al., 2015; Vey et al., 2015).

**Table 2.** Summary of the turbine specifics.

| | |
|---|---|
| Tower height | 2.1m |
| Tower diameter | 0.273m |
| Rotor diameter | 3.0m |
| Rotor overhang | 0.5m |
| Rotor blade airfoil | Clark-Y |
| Rated RPM | $180min^{-1}$ |
| Inflow velocity | $6.5ms^{-1}$ |
| TSR | 4.35 |
| 3-hole probe position | $65\%R$, $75\%R$, $85\%R$ |
| Reynolds number ($75\%R$) | 265000 |

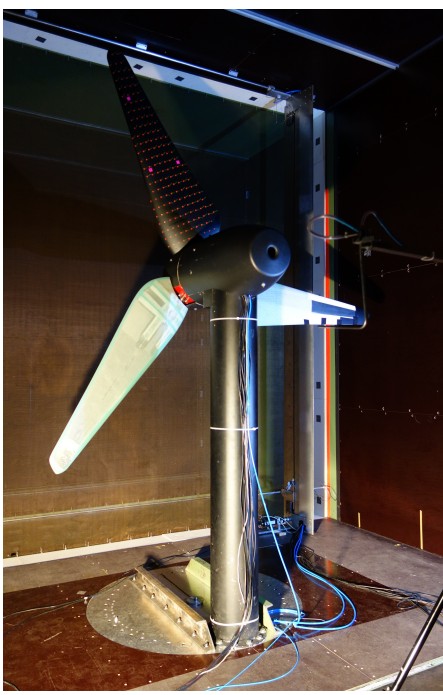

**Figure 3.** The model wind turbine *BeRT* in the wind tunnel.

The model creates a significant level of blockage of $\beta = A_{BeRT}/A_{tunnel} = 40\%$. This value is far beyond blockage ratios where correction methods have proven their applicability. But as one of the aims of the present study is the comparison between experiment and simulation, and not to quantify the overall performance to a turbine in the far field, the high blockage has only a small impact on the validity of the results.

5 Data acquisition is achieved by *National Instrument* hardware in the rotating and in the non-rotating system. In the former, a *cRIO* 9068 platform with 9220 modules rotates with the turbine and acquires data from sensors placed on the blades. In the non-rotating setup, a *National Instruments cDAQ* 9188 with 9220 modules platform collects data from additional sensors, such as tower / nacelle acceleration and tower base strain for thrust measurements. Data transmission between the two systems and the control computer is achieved by *WiFi* connection. Further information on the setup is found in (Vey et al., 2015).

### 2.2.3 Blades

The turbine is equipped with two baseline blades and one smart blade. The smart blade is equipped with a multitude of sensors and actuators for trailing edge flap deployment, whereas one of the baseline blades is equipped with blade root bending sensors. Besides that, no other sensors or actuators are mounted on the baseline blades (Bartholomay et al., 2017).

15 The smart blade, Fig. 4, is equipped with pressure ports, strain gauges at the blade root, acceleration sensors at the tip, 3-hole

probes to measure the angle of attack at $65\%R$, $75\%R$ and $85\%R$, trailing edge flap actuators and encoders to measure the flap position. The pressure sensors are *Sensortechnics HCL0075E* and the blade strain gauges are of type *FAET-A6194-N-35-S6/EL*. For the current study, the flaps were not deflected but fixed in their neutral position (Bartholomay et al., 2017). The three-hole probes, their holder and tubing change the flow around the blade. The equipment is positioned on the pressure side, as in contrast to the suction side, this side is less prone to separation. It is assumed that the presence of the installation leads to higher camber and therefore a higher local lift. Nonetheless, the installation of multi-hole probes is a common practice on research turbines, see Castaignet et al. (2014); Gallant and Johnson (2016); Pedersen et al. (2017). The strain-gauges for the determination of the blade root bending moments are glued on the bolt, Fig. 4, that connects the blades to the hub. The full-bridge aims to mitigate cross-talk effects that influence the measurement results. Nonetheless, as positioning the strain gauges on the circular bolt is challenging, cross-talk effects are present on the results of the sensors. The main sources of cross-talk are edgewise bending moments on the flapwise sensor and vice versa, axial forces due to weight and centrifugal acceleration, but they can also be caused by the blade twist. The first two effects can be quantified by calibration and compensated for measurements.

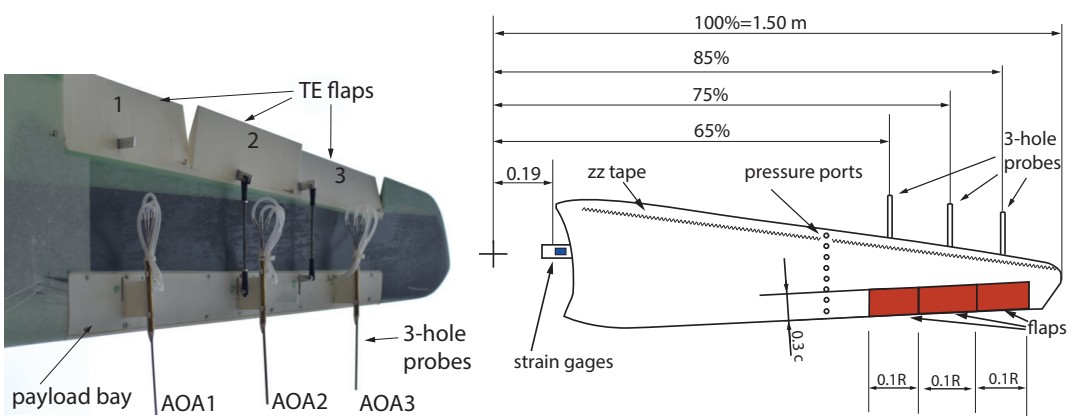

**Figure 4.** Smart blade, modified from (Bartholomay et al., 2017).

## 2.3 The Lifting Line Free Vortex Wake Code *QBlade*

The next two parts describe the numerical methods of *QBlade* and give some information about the numerical setup.

### 2.3.1 Numerical methods of *QBlade*

The Lifting Line Free Vortex Wake (*LLFVW*) computations in this study are performed with the wind turbine design and simulation tool *QBlade* (Marten et al., 2010, 2016, 2015), which is developed at the Technical University of Berlin. The *LLFVW* algorithm is loosely based on the non-linear lifting line formulation as described by Van Garrel (2003) and its implementation in *QBlade* is used to simulate both HAWT and VAWT rotors.

Rotor forces are evaluated on a blade element basis from tabulated lift and drag polar data. The wake is modelled with vortex line elements, which are shed at the blades trailing edge during every time step and then undergo free convection behind the rotor. Vortex elements are de-singularized using a cut off method, as described by Marten et al. (2016), based on the vortex core size. Viscous diffusion in the wake is accounted for through vortex core growths term.

The tower shadow is taken into account by using a model derived from the work of Bak et al. (2001), in which the tower is modelled through a combination of the analytical potential flow around a cylinder superimposed with an empirical downwind wake model based on a tower drag coefficient.

The effects of unsteady aerodynamics and dynamic stall are introduced via the *ATEFlap* aerodynamic model. This model reconstructs lift and drag hysteresis curves from a decomposition of the lift polars and has been adapted to be implemented into

the free vortex wake formulation of *QBlade*, see Wendler et al. (2016). The computational efficiency of the *LLFVW* calculations is increased through a GPU parallelization of the wake convection step via the OpenCL framework.

### 2.3.2   Numerical setup of *QBlade*

As it is currently not possible to include the wind tunnel walls into the *LLFVW* simulations of *QBlade*, far field simulations

were conducted.

The lift and drag polar data for the rotor's *Clark-Y* airfoil is obtained through *XFOIL* (Drela and GILES, 1987) calculations ($NCrit = 9$ and forced transition at leading edge) for a range of Reynolds numbers and then extrapolated to $360°$ angles of attack using the Montgomerie method (Montgomerie, 2004). Although there are similarities between the Lifting Line Free Vortex Wake method and the Blade Element Momentum Theory (*BEM*), the *LLFVW* has a main advantage when compared

to *BEM* codes. This advantage comes from the calculation of the induction from the three dimensional representation of the wake. In this representation the calculation of induction is not limited to an annular averaged rotor disc, but can be accurately calculated at any point in the computational domain and any point in time. In addition to that, the wake always contains the history of the flow (through vortex elements from previous time steps) which gives the ability to simulate transient events with a much higher accuracy than the *BEM*. Furthermore, other induction related effects such as blade hub and tip losses are directly

modelled in this formulation. Effects such as yaw error, wake memory, transient or sheared inflow are directly included in the *LLFVW* through the explicit calculation of the wake evolution in three dimensions. Overall the *LLFVW* method relies on far less semi-empirical corrections than the *BEM* when the operating conditions deviate from idealized uniform steady state inflow conditions. And thus it produces results with increased accuracy for a range of operating conditions. The advantages of vortex codes over traditional *BEM* methods, especially in unsteady operating conditions, have already been presented in numerous

publications such as Marten et al. (2016); Saverin et al. (2016a, b).

The main simulation parameters used in the *LLFVW* simulation of this study are given in Table 3.

The azimuthal discretization of $5°$ was chosen to achieve a compromise between computational efficiency and accuracy. The wake was fully resolved for eight revolutions to obtain high quality results in rotor plane region, after which it was truncated. This means, that a wake element is removed from the domain after the rotor completes eight full revolutions after it has been

**Table 3.** Main parameters of the *QBlade* simulations.

| | |
|---|---|
| Azimuthal discretization | 5° |
| Blade discretization | 21 (sinusoidal spacing) |
| Maximum wake length | $8rev$ |
| Simulation length | $16rev$ |
| Initial vortex core size | 0.025m |
| Turbulent vortex viscosity | 50 |

released from the blades trailing edge. The blade was discretized into 21 panels in radial direction using sinusoidal spacing to obtain a higher resolution in the tip and hub regions where the largest gradients in circulation are expected. The simulation was carried out over 16 revolutions resulting in 1152 time steps and a maximum of $52,000$ wake segments. Fig. 5 shows a snapshot of the *LLFVW* simulation after four rotor revolutions.

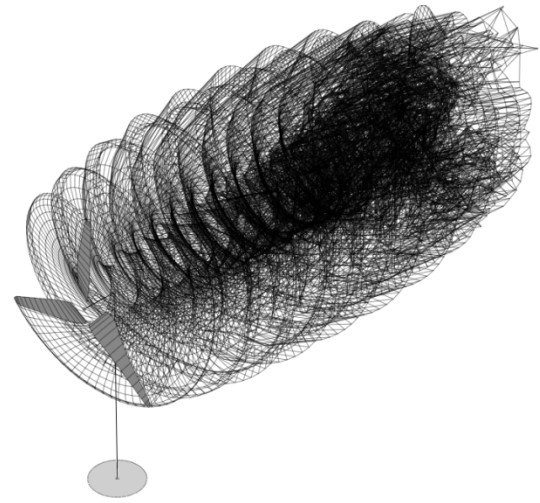

**Figure 5.** Snapshot of the *LLFVW* simulation after four rotor revolutions.

## 2.4 The CFD Code *FLOWer*

In the following, general information about *FLOWer* are given. Moreover, information about the numerical FLOWer setup are provided.

### 2.4.1 Numerical methods of *FLOWer*

The *URANS* simulations are carried out using the block-structured solver *FLOWer*, which uses the finite volume method. It solves the compressible Navier-Stokes-Equations and was developed by the German Aerospace Center (*DLR*) in the course of the *MEGAFLOW* project (Kroll et al., 2000) whereas wind energy specific extensions were made at the Institute of Aerodynamics and Gas Dynamics (*IAG*) of the University of Stuttgart. For the temporal discretization, an implicit dual time stepping scheme is used (Jameson, 1991). The space is discretized with a second order central discretization scheme *JST* (Jameson et al., 1981). For the modelling of the turbulence, the Menter *SST* turbulence model is used and the simulations are performed fully turbulent. All components of the setup are meshed separately with a fully resolved boundary layer ($y^+ \approx 1$) and all grids are overlapped, using the *CHIMERA* technique (Benek et al., 1986). The process chain, as used for the present investigations, was developed at the *IAG* (Meister, 2015).

### 2.4.2 Numerical setup of *FLOWer*

The numerical setup consists of eleven grids: background grid (wind tunnel WT or far field FF), hub, nacelle, 3×connection for the blade (blade con), 3×blade, tower and connection for the tower (tower con). The number of cells per grid for all cases can be found in Table 4.

Altogether, the setup in the wind tunnel has 40.1 mio cells. In the far field case, where the wind tunnel walls are not modelled

**Table 4.** Cell number in mio of the individual grids for the wind tunnel and far field cases.

| Wind tunnel (WT) | | | | | | | |
|---|---|---|---|---|---|---|---|
| Name | WT | Hub | Nacelle | Blade con | Blade | Tower con | Tower |
| No. of cells [mio] | 11.7 | 2.2 | 1.3 | 0.5 | 7.2 | 0.2 | 1.6 |
| Far field (FF) | | | | | | | |
| Name | FF | Hub | Nacelle | Blade con | Blade | Tower con | Tower |
| No. of cells [mio] | 14.7 | 2.2 | 1.3 | 0.5 | 5.5 | 0.2 | 1.6 |

and the background grid has a large expansion, the setup features 38.0 mio cells.

The blade is meshed automatically and is of CH-topology. The boundary layer is fully resolved with 37 grid layers, ensuring $y^+ < 1$ for the first grid layer. Around the airfoil 181 cells were used, in spanwise direction 145 cells for the wind tunnel case and 101 for the far field case. For the wind tunnel case, at around $60\%$ of the radius and at around $90\%$ of the radius, spanwise refinements were introduced, which ensure a proper transition for future trailing edge flap deflection. The meshes for all other components, except the far field mesh, are created manually.

Klein et al. (2018) already showed that the wind tunnel walls, the tower and the nozzle behind the turbine have a significant influence on the turbine performance. Therefore, they are taken into account for the present *CFD* simulations. The $4.2 \times 4.2 \mathrm{m}^2$

settling chamber of the *GroWiKa* begins 1.245m upstream of the rotor plane and is 5.0m long. As the original test section of the wind tunnel is located behind the settling chamber, in this configuration, the nozzle is located behind the "new" test section. It has a total length of 3.0m and a tapering of 2.2. The wind tunnel walls are realized as slip walls, whereby an approximated displacement thickness, based on the turbulent flow over a flat plate, is added on the real walls. This leads to a constant reduc-

tion of the cross section over the whole settling chamber.

In order to prevent the convection of disturbances from the inflow and outflow planes of the computational domain into the measuring section, the wind tunnel was extended to a length of approximately $16.5R$, whereas the rotor plane is located after approximately $7.5R$. The cells around the turbine have an extension of $0.025 \times 0.025 \times 0.025 \text{m}^3$. In the direction of the inflow, the cells are stretched up to 0.4m in x-direction, at the outflow, they measure $0.2 \times 0.025 \times 0.025 \text{m}^3$. The inflow boundary is

realized as far field and at the outflow, a constant pressure is defined in order to maintain mass continuity.

As the wind tunnel and the nozzle could not be taken into account in *QBlade*, yet, a far field case was created, too. Thereby, the refinement for the flaps in the blade mesh was not realized. The background mesh for the far field case was created by an automated script (Kowarsch et al., 2016), which uses hanging grid nodes for the refinement. Usually, in a H-topology, the refinement is not only at the designated spot, but has to be taken along to unnecessary areas. With hanging grid nodes,

refinements can be realized only where they are needed. The grid has an overall length of $20.5R$ ($8R$ upstream and $12.5R$ downstream of the rotor), a width of approximately $24.6R$ and a height of approximately $14R$. Consequently, the boundaries are, according to Sayed et al. (2015), far away enough to prevent disturbances on the solution. The boundaries, except the bottom, which is realized as slip-wall, are realized as far field boundary condition. Around the turbine, the cells have a dimension of $0.025 \times 0.025 \times 0.025 \text{m}^3$, at the borders $0.1 \times 0.1 \times 0.1 \text{m}^3$.

For a one third model a grid convergence index study according to Celik et al. (2008) was already performed (Fischer et al., 2018). The extrapolated relative errors between the appropriate grids and the extrapolated values of a theoretical ideal mesh, which were determined in the course of this investigation, amount $0.63\%$ for power and $0.02\%$ for thrust. As the grids used for the present investigation are partly finer resolved than the ones used in the sensitivity analysis, a renewed investigation for the full model was not performed. As the cell number is limited in the numerical simulation and the modelling effort is significant,

measuring equipment in the wind tunnel and on the blades was not taken into account.

For the wind tunnel cases, the simulations were performed until convergence of the loads was achieved. This occurs when the difference between the average of torque and thrust over five revolutions and the average of the following five revolutions is $< 0.1\%$. Afterwards, the average of the last five revolutions were used for the evaluation. For the present investigation, 45 rotor revolutions were calculated in total. The temporal discretization corresponds to $1.5°$ azimuth and 100 inner iterations for the

cases including wind tunnel walls and $1.5°$ azimuth with 30 inner iterations for the far field case.

## 3   Data acquisition

This section deals with the data acquisition of the velocity planes, the on-blade velocity and angle of attack as well as of the bending moments for each the experiment and the simulations. Fig. 6 shows the position of the velocity planes as well as the

evaluation surfaces for the *CircAve* (*LineAve* with circles) method for the AoA determination in *FLOWer* (see subsection 3.2) exemplary at blade 1 and the surfaces used for the *RAV* method of the AoA determination in *FLOWer* (see subsection 3.2).

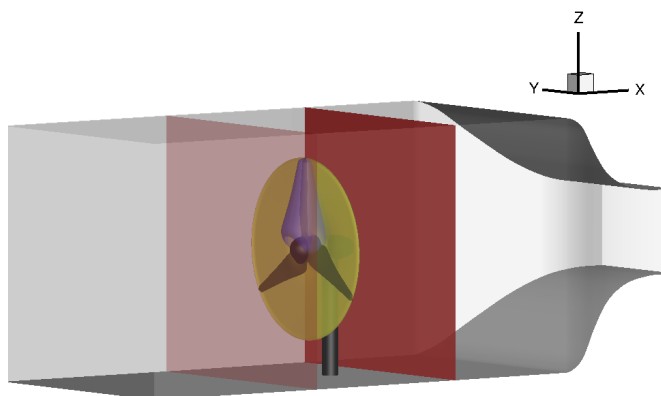

**Figure 6.** Position of the velocities planes for the *RAV* method (yellow), surface for the determination of the AoA with the *CircAve* method (blue) and velocity planes (red).

## 3.1   Generation of the velocity planes

In the experiment, the three red dots in Fig. 2 (left) at $x = -0.43d$, $x = 0.5d$ and $x = 1.05d$ indicate where hot-wire measure-
ments are conducted. A semi-automatic traverse with four cross-wire probes with a measurement frequency of $f_s = 25\text{kHz}$ and a cut-off frequency of $f_{cut} = 10\text{kHz}$ is used. Each of the 608 measurement positions, Fig. 2 (right), in each cross-section is measured for $T_s = 16\text{s}$. This time is assumed to be long enough for good statistics for the current setting as the measured integral time scale is $\leq 0.023\text{s}$, which is considerably smaller than the acquisition time of $16\text{s}$. With the inflow velocity of $6.5\text{ms}^{-1}$ as convective velocity, an integral length of $0.023\text{s} \cdot 6.5\text{ms}^{-1} = 0.15\text{m}$ is calculated based on Taylor's hypothesis.
Offset correction between the probes was realized by repeating 19 measurement points along a vertical line with all four probes. For each measurement position, the mean value of all four measurements was calculated and used as reference. Subsequently, the offset of each probe was calculated. This offset was averaged over all measurement points. Thereby, the offset for each probe was calculated, which was then applied to all measurements in the post-processing. The calibration of the probes was done with the help of a nearby pitot-probe at different wind tunnel velocities.
The error of the hot-wire measurements is the sum of the calibration setup error (pitot-tube, pressure sensor) and the hot-wire anemometry hardware. The latter was calculated by measuring multiple points in each test case with all probes and the largest deviation is defined as the error. In the present case it amounts $3.3\%$, which corresponds to $0.33\text{ms}^{-1}$ in reference to the maximum calibrated velocity. This is in good agreement with error estimations given in literature, see Finn (2002). The total error, including calibration setup, is calculated to $4.4\%$, corresponding to $0.44\text{ms}^{-1}$.
Only the simulation including wind tunnel walls has been taken into account for the comparison of the velocity planes. In this setup, at each point of the numerical grid, data was extracted for the planes and averaged over five revolutions. In order to

evaluate the differences between measurement and simulation, the results of the simulation are interpolated to a grid with the same grid points as the measurement points and the results are subtracted.

## 3.2 Extraction of the on-blade velocity and the angle of attack

The angle of attack (AoA) is the angle between the velocity, as seen by the blade (on-blade velocity), and the airfoil chord. Generally, deriving an angle of attack in rotating domain is somewhat difficult, as the AoA is a two-dimensional value. Moreover, the blade deflects the streamtraces due to its induction and therefore changes the value of the AoA.

In the experiment the AoA and the on-blade velocity are measured by three-hole probes located at $65\%R$ and $85\%R$. The derivation of the section-wise values, referenced to the quarter-chord point of each section, is detailed by Bartholomay et al.
(2017) and will be explained here shortly. Generally, this measurement method is advantageous, as no static tunnel reference pressure is needed and short tubing, as the pressure sensors are located in the blade, mitigates possible delay effects. The three-hole probes measure the $\alpha_{probe}$ and $U_{rel,probe}$ in reference to the probe position upstream of the wing. These values are derived by calibration of the pressure differences between tubes to the flow angle and velocity. However, when mounted on the wing, the results are affected by the induction of the blade and therefore need to be translated into the sectional angle of attack
$\alpha$ and the relative velocity $U_{rel}$. In this project a procedure based on two dimensional flow assumption on the wing, Fig. 7, was employed.

Herein, $\alpha_{probe}$ is first rotated into the local coordinate system, which is based on the local chord, to derive $\alpha_{probe,section}$.

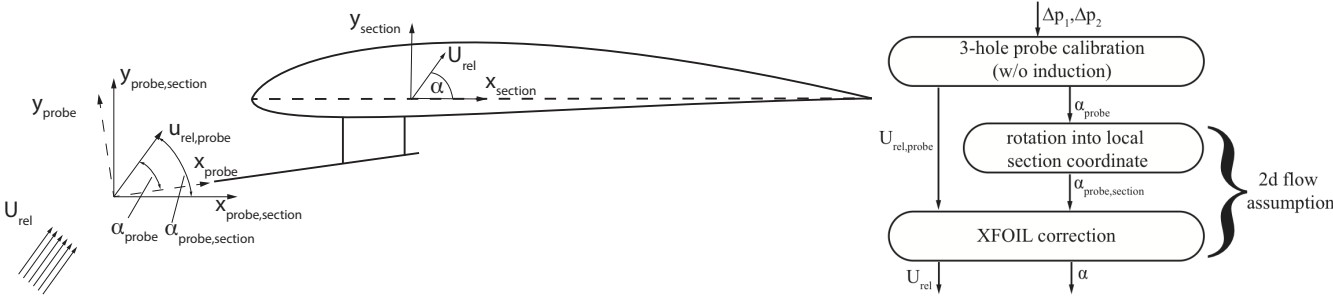

**Figure 7.** Schematic and flow chart of derivation of the section-wise AoA (Bartholomay et al., 2017).

Subsequently, a look-up table is used, that was derived by viscous *XFOIL* (Drela and Youngren, 2008) calculations. This table correlates the measurement at the probes head upstream of the wing to the actual local section angle of attack $\alpha$. Thereby, the
induction effect is accounted for and $\alpha$ and $U_{rel}$ are found. The analysis showed, that the dependency of the local flow angle at the probe to the actual AoA is almost a first order function in the linear region of the lift polar (the AoA range where the lift has a nearly constant slope). The approximated equation (Eq. 1) gives information about the order of conversion for this 2D approach.

$$\alpha = 0.58° \cdot \alpha_{probe} - 0.64° \tag{1}$$

The data-set was created by analyzing polars from $\alpha = -30°$ to $30°$ in steps of $0.5°$. Steps in-between are interpolated. This procedure requires two-dimensional flow over the blade, which is assumed to be appropriate in this case, in comparison to quantitative tuft flow analysis (Vey et al., 2015), which indicated little three-dimensional effects on the surface flow.

In order to estimate the measurement error of the three-hole probes, data sets from calibrations of the probe alone and of measurements of the probe installed in a 2D-wing setup were analyzed. The data sets include variation of AoA from $-30°$ to $30°$ and the variation of the free stream velocity. From this analysis, which also includes the error of the induction correction and sensor uncertainties, the maximal absolute error for AoA was estimated to be $0.8°$ (considering only the attached flow regime) and for the on-blade velocity to be $0.4\mathrm{ms}^{-1}$.

In *QBlade*, the angles of attack are evaluated at the quarter chord position of the airfoils at the lifting line (the bound vorticity) of the rotor blades. The angle of attack is calculated from the part of the absolute velocity vector that lies inside the respective airfoils cross sectional plane – which corresponds to the on-blade velocity. The absolute velocity vector itself is a superposition of the inflow, relative, wake-induced and self-induced velocity vectors.

Different methods to derive the effective sectional AoA from 3D *CFD* predicted flow fields are compared and evaluated by Jost et al. (2018). Details of the methods are described in that manuscript. The two methods, which are most suitable for the present case, are used for the AoA extraction shown in this paper. The reduced axial velocity method (herein after called *RAV*) uses two planes, one upstream and one downstream of the rotor (see Fig. 6). In these planes, the average velocities are calculated and afterwards the velocity components are used to determine the velocity in the rotor plane without the induction of the blade. The method bases on the method of Johansen and Sørensen (2004), who determined airfoil characteristics from 3D *CFD* rotor computations. It was successfully applied by Jost et al. (2016) to investigate unsteady 3D effects on trailing edge flaps, and by Klein et al. (2014) for *CFD* analysis of a 2-bladed multi-megawatt turbine. In the line averaging method (*LineAve* or *CircAve*), the AoA is determined by averaging the velocity over a closed line around each blade cut (see Fig. 6). For both approaches, the results are averaged over five revolutions.

## 3.3 Determination of the bending moments

In the present paper, the flapwise (out-of plane) moment ($M_y$) and the edgewise (in plane) moment ($M_x$) are investigated.

Due to problems with the full-bridge strain-gauge setup in the experiment, strong fluctuations are visible in the raw data and heavy filtering was necessary. Therefore, the bending moments can not yet be considered as valid basis for quantitative comparisons and code validation purposes.

In the *LLFVW* method of *QBlade* the blade bending moments are evaluated by summing up the elemental blade forces, obtained from an integration of the normal and tangential forces along the blade span that are obtained via the stored airfoil coefficients.

In the *CFD* simulation, the bending moments in the blade root result from the pressure and friction on the blade surface. For each surface cell the forces are computed and multiplied with the corresponding radius. Then, they are averaged over five

revolutions.

# 4 Results and discussion

## 4.1 Comparison of the velocity planes

The velocity planes, which are taken into account in the present study, are placed $0.43d$ upstream and $0.5d$ downstream of the rotor plane (see Fig. 6). The plane $1.05d$ downstream of the rotor plane (see Fig. 2), is neglected in the present study, as the evaluation would not have brought further benefit for the paper. Moreover, at this location, the influence of the nozzle is already present, which influences the wake development on top of the wind tunnel walls.

Fig. 8 (left) shows the velocity in x-direction for the measurement and the right picture for the *FLOWer* wind tunnel simulation $0.43d$ upstream of the rotor plane. The measuring points are shown as black dots. The dimensions of the wind tunnel, as well as the model wind turbine, are illustrated by dashed lines. Moreover, an isoline with the undisturbed inflow velocity of $6.5\mathrm{ms}^{-1}$ is shown. The view direction in this picture, and in all following figures of the velocity planes, is from downstream to upstream. The turbine blockage effect can be observed in both figures. However, the velocity distribution in the simulation is smoother

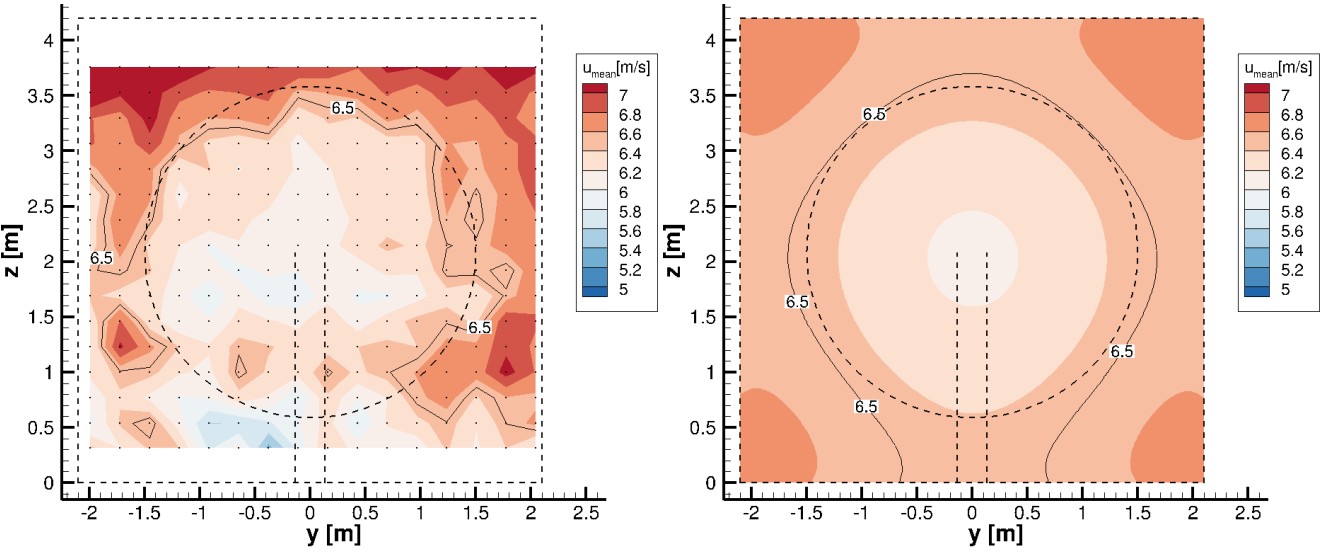

**Figure 8.** : Hot-wire measurements (left) and simulated velocity plane (right) of the x-velocity $0.43d$ upstream of the rotor plane. The dashed lines illustrate the wind tunnel and the turbine. Isolines show the undisturbed inflow velocity of $6.5\mathrm{ms}^{-1}$. The dots in the left figure show the discrete measuring points.

and axisymmetric, leading to a clearly defined blockage, whereas it is more frayed in the experiment. Due to the location of the settling chamber after a corner, see Fig. 2, the measured x-velocity on the left side differs slightly to the velocity on the right side. Additionally, a difference at the bottom and upper position is apparent. As due to constructive reasons, the mounting

of the aforementioned filtermat (see subsection 2.2.1) leaves a small gap at the ceiling of the wind tunnel, a small velocity overshoot is present at the top of the inflow test-section. In the simulation, a slightly higher velocity can be seen in the corners of the wind tunnel.

In the experiment, multiple causes of possible measurement errors, such as temperature compensation or induction of the traversing system are analyzed and ruled out. Therefore, the horizontal inequalities seem to result from the design of the wind tunnel. More information about the hot wire measurements and possible reasons for the inequality of the flow field can be found in Bartholomay et al. (2017).

Table 5 gives an overview of some mean parameters characterizing the velocity plane $0.43d$ upstream of the rotor plane. In the experiment, the averaging was done over the measuring time, in the simulation over five revolutions. The mean velocities in

**Table 5.** Mean parameters for the velocity plane $0.43d$ upstream of the rotor plane.

|  | $\overline{u}$ [ms$^{-1}$] | $\overline{\sigma_u}$ [ms$^{-1}$] | $\overline{Ti}_{global}(uv)$ [%] |
|---|---|---|---|
| Measurement | 6.42 | $8.50 \cdot 10^{-2}$ | 1.20 |
| *FLOWer* | 6.47 | — | — |

streamwise direction are slightly smaller than the desired velocity, both for measurement and simulation. However, as the differences are $< 0.5\%$ in the simulation and $\approx 1\%$ in the measurement, the reference velocity can still be considered as $6.5\text{ms}^{-1}$. As uniform inflow was used in the present simulation, the standard deviation and turbulence intensity are negligible. The turbulence intensity of the measurement corresponds to the value of the wind tunnel, which was already mentioned in subsection 2.2.1. The unsteady inflow in the experiment and the uniform inflow in the simulation lead to a discrepancy in the setups. The influence of the turbulence on the results will be discussed later in this document and reviewed in future investigations.

In Fig. 9, the relative difference between simulation and measurement with regard to the mean inflow velocity of $6.5\text{ms}^{-1}$ is shown.

The differences between both velocity planes are small as the average deviation amounts $\approx 3\%$. Except for a small area at the bottom of the wind tunnel (around $z = 0.5\text{m}$ and between $-1\text{m} < y < 0\text{m}$), the difference is lower than $\pm 10\%$ of the desired inflow velocity, which corresponds to $\pm 0.65\text{ms}^{-1}$.

Fig. 10 shows the velocity in x-direction $0.5d$ downstream of the rotor plane, for the measurement (left) and for the simulation (right). Again, the measuring points are indicated by black dots, the dimensions of the wind tunnel and the model wind turbine by dashed lines. An isoline with the mean velocity of $6.5\text{ms}^{-1}$ is shown, too.

Some aspects, as already seen upstream of the rotor (Fig. 8) are apparent downstream of the rotor, too. For example the higher velocity over the ceiling in the measurement. Or the smoother, axisymmetric streamwise velocity in the simulation. In both figures (left and right), the wake of the rotor, indicated by lower velocity, can be seen clearly. Around the rotor, as a result of limited space due to the wind tunnel walls, higher velocities are achieved. Again, in the experiment, the velocity at the upper part of the wind tunnel is slightly higher than at the bottom.

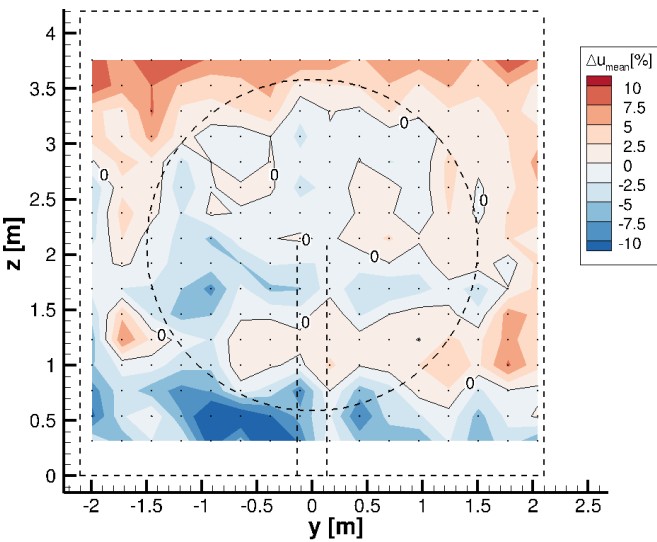

**Figure 9.** Relative velocity difference between measurement and simulation with regard to the undisturbed reference inflow velocity of $6.5\mathrm{ms}^{-1}$, $0.43d$ upstream of the rotor plane. The dashed lines illustrate the wind tunnel and the turbine. Isolines show $0\%$ deviation. The dots show the discrete evaluation points.

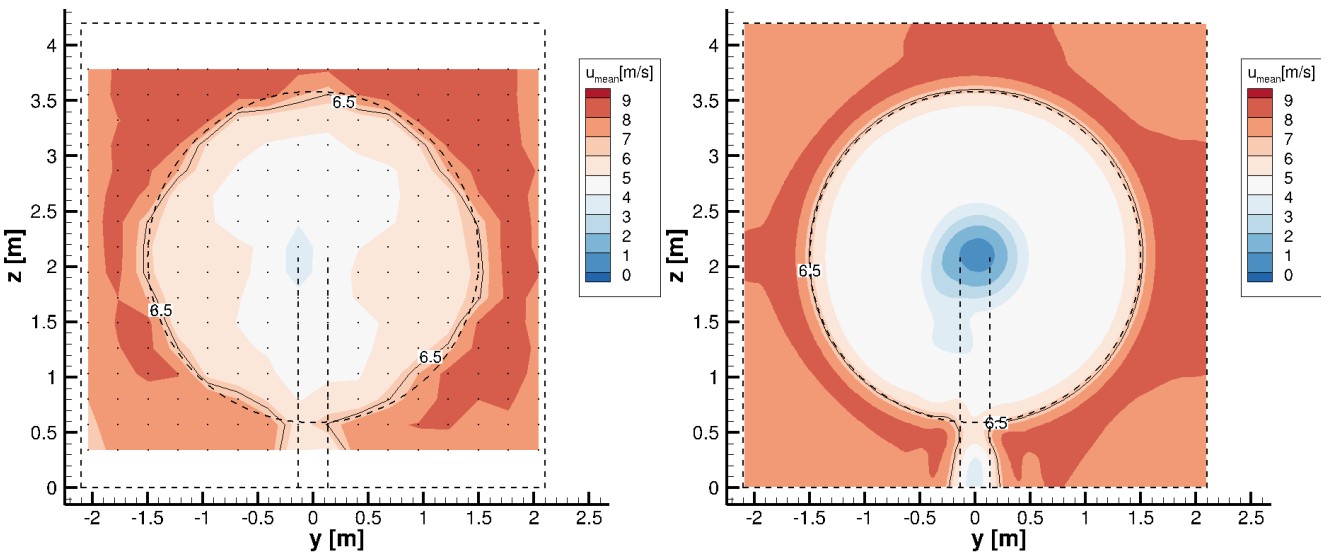

**Figure 10.** Hot-wire measurements (left) and simulated velocity plane (right) of the x-velocity $0.5d$ downstream of the rotor plane. The dashed lines illustrate the wind tunnel and the turbine. Isolines show the mean inflow velocity of $6.5\mathrm{ms}^{-1}$. The dots in the left figure show the discrete measuring points.

This missing turbulence in the simulated wind tunnel is the reason why the border of the rotor wake is almost a perfect circle in the right picture, whereas it is more smeared in the measurement. The decay of the tip vortices has not yet started so shortly

behind the rotor plane. As the simulation has a finer resolution, the velocity distribution is smoother there. In the simulation, there is a stronger velocity deficit in the wake of the nacelle. This can have several reasons. In the simulation, the missing inflow turbulence might have a small effect on the stability of the wake, but is certainly not the main reason for the deviation, see Medici and Alfredsson (2006). In the experiment, the boundary layer of the nacelle is not tripped, whereas a fully turbulent

approach is used in the simulation. These differences concerning the boundary layer of the nacelle might lead to a different recovery of the wake of the nacelle. Due to the flow separation on the nacelle, the flow in the wake of the nacelle is highly unsteady and the main flow direction is not clearly defined (angles larger than $\pm 60°$ occur in the simulation), whereby proper working conditions of the x-wire probe are no longer guaranteed. Therefore, the measured x-component of the velocity is influenced by the y- and z-component, which could also lead to deviations between measurement and simulation.

An overview of some mean parameters characterizing the velocity plane $0.5d$ downstream of the rotor plane are given in Table 6.

Again, the mean velocity almost corresponds to the desired reference velocity, as the differences between the actual velocity

**Table 6.** Mean parameters for the velocity plane $0.5d$ downstream of the rotor plane.

|  | $\overline{u}$ [ms$^{-1}$] | $\overline{\sigma_u}$ [ms$^{-1}$] | $\overline{Ti}_{global}(uv)$ [%] |
| --- | --- | --- | --- |
| Measurement | 6.53 | $6.76 \cdot 10^{-1}$ | 7.01 |
| *FLOWer* | 6.48 | $3.17 \cdot 10^{-1}$ | 3.71 |

and $6.5\text{ms}^{-1}$ are $< 0.5\%$ both for measurement and simulation. Due to the closed wind tunnel and the mass continuity, bigger differences would not have been physical. As the tip and root vortices, as well as the separation behind the nacelle, lead to

velocity fluctuations, the standard deviation, as well as the turbulence intensity, increase compared to the plane upstream to the rotor, see Table 5. Through the superposition of the vortices created by the turbine and the inflow turbulence, the values for the measurement are still larger. As the present wind tunnel is a circuit wind tunnel, effects like pumping might occur. And due to the long measurement time of the hot wire probes, these fluctuations might also be included in the values shown in Table 6. Fig. 11 shows the relative difference between simulation and measurement with regard to the mean inflow velocity of $6.5\text{ms}^{-1}$.

It can be seen that in the wake of the nacelle and in the area of the tip vortices, the differences between simulation and mea- surement are higher that $10\%$. In the remaining part, the difference is smaller. The mean deviation amounts $\approx 7\%$, which is considerably higher than the value for the plane upstream of the turbine. The reason for the high value is primary the area in the wake of the nacelle, where differences $> 50\%$ occur. If a circular area with a radius $r < 0.56\text{m}$ and its origin at the center of the rotor is neglected in the averaging, the mean deviation reduces to $< 6\%$ as the mean deviation in this area itself amounts

about $31\%$. Thereby it has to be kept in mind, that due to the large flow angles in the wake of the nacelle, the measured values in this area have to be treated with caution.

All things considered, the accordance between experiment and simulation is acceptable, as the differences are, except for some

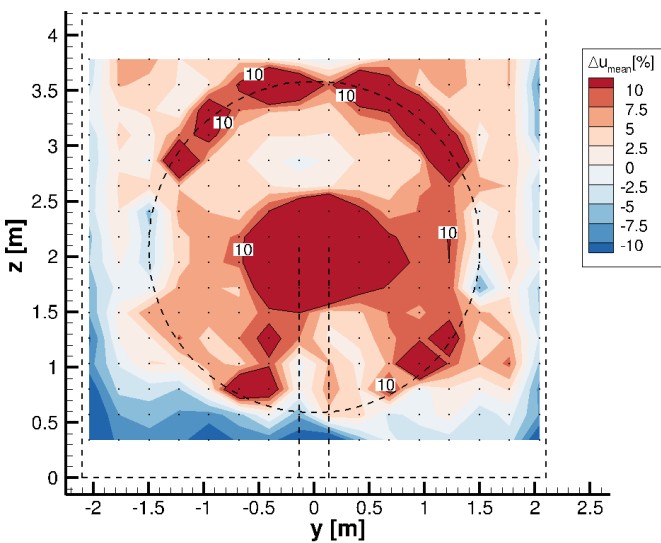

**Figure 11.** Relative velocity difference between measurement and simulation with regard to the undisturbed reference inflow velocity of $6.5\mathrm{ms}^{-1}$, $0.5d$ downstream of the rotor plane. The dashed lines illustrate the wind tunnel and the turbine. Isolines show $10\%$ deviation. The dots show the discrete evaluation points.

parts in the outer region of the rotor and in the wake of the nacelle, smaller than $\pm 0.65\mathrm{ms}^{-1}$.

## 4.2 Analysis of the on-blade velocity

Hereinafter, the on-blade velocity, meaning the velocity seen by the blade section at a distinct radial position, for $Case\,BASE$
for experiment, *QBlade* and *FLOWer* (both methods *RAV* and *CircAve*) are displayed at two different rotor locations ($65\%R$
and $85\%R$) over the azimuth (Fig. 12). A radius of $0\%R$ corresponds to the rotor center, whereas an azimuth of $0°$ corresponds
to the top position of the first blade .

At $65\%R$, the simulations overestimate the velocity, at $85\%R$ there is a better accordance between the simulation results
and the experiment. The difference caused by the different inflow turbulence is even less pronounced at the on-blade velocity
compared to the velocity planes, as the rotational velocity has a much higher influence than the inflow velocity. Therefore,
the fluctuations in the measurements are not so distinct and the differences between measurement and simulation caused by
the inflow turbulence are small. For their cases with and without free stream turbulence, Medici and Alfredsson (2006) also
experienced only small differences in the drag coefficient, which depends on the angle of attack and consequently also on the
on-blade velocity. The higher fluctuations in the experiment at the outer radial position might be a result of a vibration of the
mounting of the probe. The averaged standard deviation for the measured velocity amounts $\sigma_{on-blade}(65\%R) = 0.11\mathrm{ms}^{-1}$
and $\sigma_{on-blade}(85\%R) = 0.08\mathrm{ms}^{-1}$.

In order to better assess the quantitative differences between the curves, Table 7 gives an overview of the relative differences

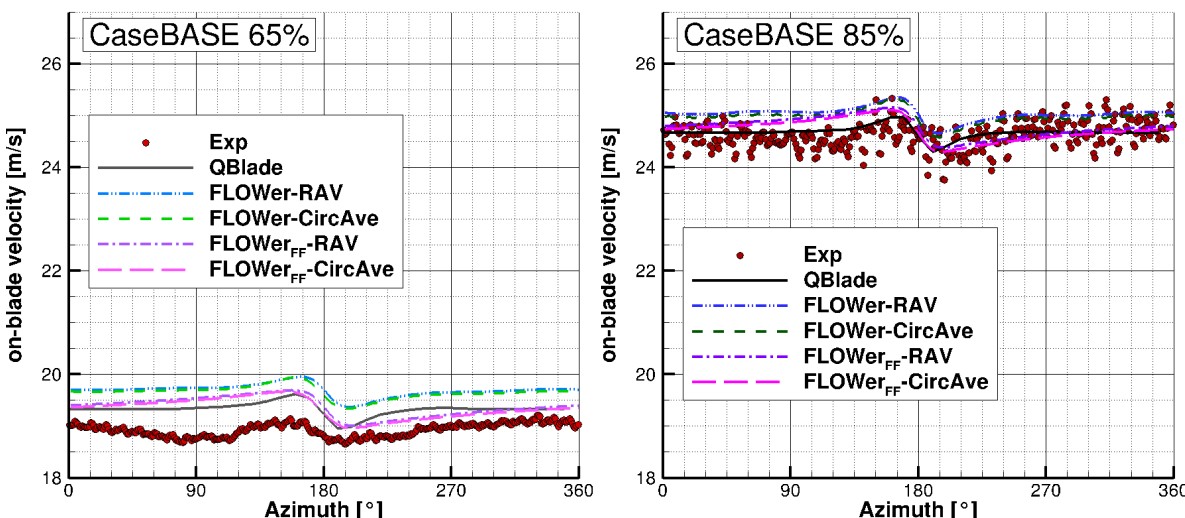

**Figure 12.** On-blade velocity distribution over azimuth for $CaseBASE$ for the experiment, *QBlade* and *FLOWer* (*RAV* and *CircAve* for wind tunnel and FF each) at $65\%R$ (left) and $85\%R$ (right).

between experiment and the different simulation results of the averaged on-blade velocity ($\Delta \overline{v} = \overline{v_{Sim}} - \overline{v_{Exp}}$) for $CaseBASE$ at both probe positions.

The reference velocity in each case is the undisturbed velocity at the probe position, which was calculated with

$$v_{Ref} = \sqrt{v_{inflow}^2 + (\omega \cdot R)^2}. \tag{2}$$

5  and amounts $v_{Ref}(65\%R) = 19.49\mathrm{ms}^{-1}$, respectively $v_{Ref}(85\%R) = 24.90\mathrm{ms}^{-1}$.

For both radial positions, all simulations match fairly well to each other, as the differences to the experiment are relatively

**Table 7.** Relative differences between the experiment and the different simulation results of the averaged on-blade velocity with respect to the undisturbed velocity at the probe positions for $CaseBASE$.

| $\Delta\overline{v}$ [%] | $QBlade$ | $FLOWer - RAV$ | $FLOWer - CircAve$ | $FLOWer_{FF} - RAV$ | $FLOWer_{FF} - CircAve$ |
|---|---|---|---|---|---|
| $65\%R$ | 2.05 | 3.90 | 3.72 | 2.31 | 2.10 |
| $85\%R$ | 0.25 | 1.68 | 1.44 | 0.68 | 0.43 |

similar. However, all simulations overestimate the experimental results. For the *FLOWer* simulations, both methods (*RAV* and *CircAve*) show almost the same results ($\Delta\overline{v}_{FLOWer-RAV}$ and $\Delta\overline{v}_{FLOWer-CircAve} \approx 4\%$ at $65\%R$ and $\Delta\overline{v}_{FLOWer-RAV}$ and $\Delta\overline{v}_{FLOWer-CircAve} < 2\%$ at $85\%R$), whereby the CircAve method seems to fit better to the experimental results. In the

10  outer part of the blade, where the probes are located, the on-blade velocity is dominated by the tangential velocity. Conse-

quently, both *FLOWer* setups (wind tunnel and far field), show almost the same results, too. But due to the wind tunnel walls, the inflow velocity in the rotor plane is slightly higher than in the far field case, which can be seen in the marginal higher curves for the wind tunnel case.

With increasing radius, the difference between the wind tunnel and the far field case decreases, as the rotational part of the
velocity becomes more and more dominant. The *QBlade* results are closest to the measured data, which is surprising, as the wind tunnel walls are not taken into account in the *LLFVW* simulations. Due to the lack of the walls, they have a better accordance with the *FLOWer* far field results than with the ones including the walls. The influence of the tower blockage around an azimuth of $180°$ can be seen at both radial positions as a small increase before the tower passage and a small drop afterwards. The increase of the inflow velocity is due to the displacement effect of the tower. Directly upstream of the tower, the velocity is
reduced until it has recovered shortly afterwards. Except for this drop, the velocity is almost constant over the whole revolution. Fig. 13 shows the velocity over azimuth under yaw=$-15°$. As the wind tunnel walls should not be neglected in the present setup, a far field case under yawed condition for *FLOWer* was not simulated.

Under $15°$ yaw misalignment, the averaged standard deviation for the measured velocity is the same as for $CaseBASE$

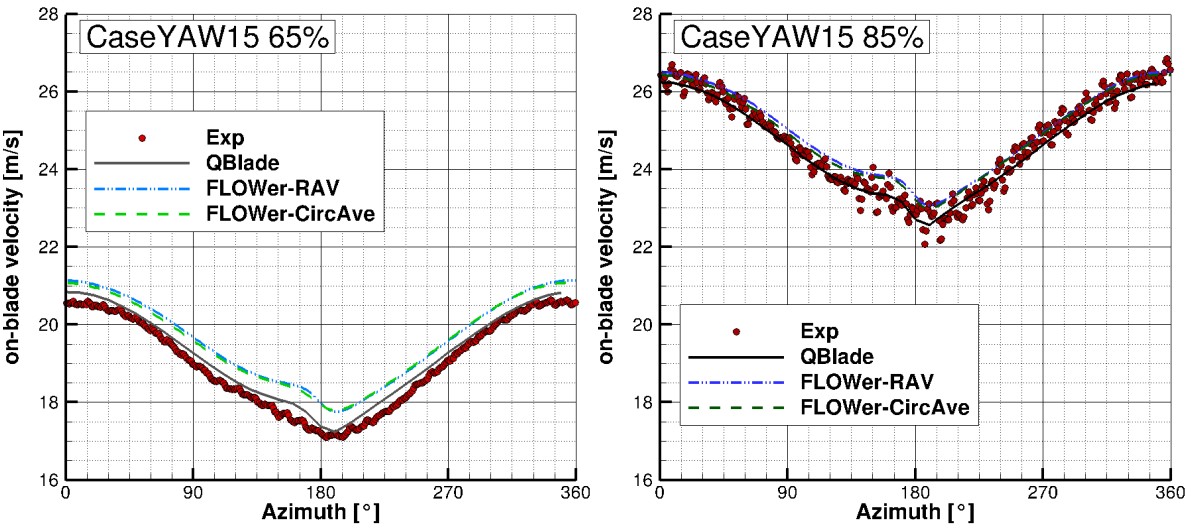

**Figure 13.** On-blade velocity distribution over azimuth for $CaseYAW15$ for the experiment, *QBlade* and *FLOWer* (*RAV* and *CircAve*) at $65\%R$ (left) and $85\%R$ (right).

($\sigma_{on-blade}(65\%R) = 0.11\text{ms}^{-1}$ and $\sigma_{on-blade}(85\%R) = 0.08\text{ms}^{-1}$). Table 8 gives an overview of the relative differences
between experiment and the different simulation results of the averaged on-blade velocity for $CaseYAW15$ at both probe positions.

At $65\%R$, the experimental and *QBlade* results are almost identical ($\Delta\overline{v}_{QBlade} \approx 1\%$), whereas *FLOWer* predicts a slightly higher velocity ($\approx 0.5\text{ms}^{-1}$, which corresponds to $\Delta\overline{v}_{FLOWer} \approx 3\%$). At $85\%R$, there is still a small offset between *QBlade* and *FLOWer*, but the measurement lies between the two curves, which can also be seen at the different signs of the differences

**Table 8.** Relative differences between the experiment and the different simulation results of the averaged on-blade velocity with respect to the undisturbed velocity at the probe positions for $CaseYAW15$.

| $\Delta\overline{v}$ [%] | $QBlade$ | $FLOWer - RAV$ | $FLOWer - CircAve$ |
|---|---|---|---|
| $65\%R$ | 0.96 | 3.04 | 2.87 |
| $85\%R$ | $-0.54$ | 1.05 | 0.82 |

in Table 8. Moreover, as already seen for the case with no yaw misalignment, the differences are smaller further outboard. In total, the differences between experiment and simulations are smaller than under straight inflow.

The influence of the tower is covered by the influence of the yaw misalignment, which leads to stronger variations over one revolution. In the upper part of the rotor (azimuth=270°-90°), the blade is advancing, while it is retreating in the lower part
5   (azimuth=90°-270°). This leads to a $1p$ variation of inflow velocity as seen by the blade. Further information and detailed discussions about effects occurring under yaw misalignment, like the $1p$ variation, are summarized by Schulz et al. (2017).

In Fig. 14, where the velocity over azimuth under yaw=$-30°$ is plotted, the influence of the yaw misalignment is even more pronounced.

Again, the averaged standard deviation for the measured velocity amounts $\sigma_{on-blade}(65\%R) = 0.11\text{ms}^{-1}$ and $\sigma_{on-blade}(85\%R) = $

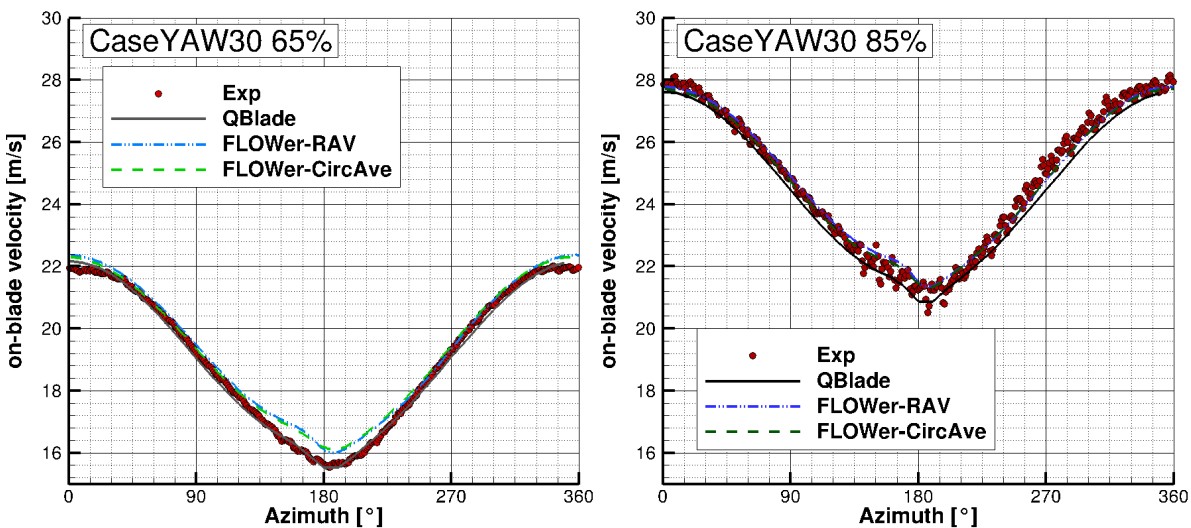

**Figure 14.** On-blade velocity distribution over azimuth for $CaseYAW30$ for the experiment, $QBlade$ and $FLOWer$ ($RAV$ and $CircAve$) at $65\%R$ (left) and $85\%R$ (right).

10   $0.08\text{ms}^{-1}$. In Table 9, the relative differences between experiment and the different simulation results of the averaged on-blade velocity for $CaseYAW30$ at both probe positions are displayed.

Almost the same characteristics as already mentioned with regard to Fig. 13 can be found for $-30°$ yaw misalignment. How-

**Table 9.** Relative differences between the experiment and the different simulation results of the averaged on-blade velocity with respect to the undisturbed velocity at the probe positions for $CaseYAW30$.

| $\Delta\overline{v}\,[\%]$ | $QBlade$ | $FLOWer-RAV$ | $FLOWer-CircAve$ |
|---|---|---|---|
| $65\%R$ | $-0.79$ | $1.41$ | $1.30$ |
| $85\%R$ | $-1.65$ | $0.11$ | $-0.1$ |

ever, at $65\%$R, the *FLOWer* results have a better agreement with the experiment in the upper part of the rotor ($270°$ to $90°$ azimuth) than in the lower part ($90°$ to $270°$ azimuth). At $85\%$R the *FLOWer* curves and the measured curve correspond well ($|\Delta\overline{v}_{FLOWer}| \leq 0.11\%$, whereas the *QBlade* results have a bigger deviation to the experimental results. Overall, the differences between the simulated curves and the measured curves decrease again with increasing yaw misalignment.

### 4.3 Evaluation of the angle of attack

As for the on-blade velocity, in the following, the AoA for $CaseBASE$ for experiment, *QBlade* and *FLOWer* (both methods *RAV* and *CircAve*) are displayed at two different rotor locations ($65\%$ and $85\%$) over the azimuth (Fig. 15).

The tower blockage effect can be clearly seen at azimuth=$180°$, where the AoA has a drop of approximately $1°$. The influence

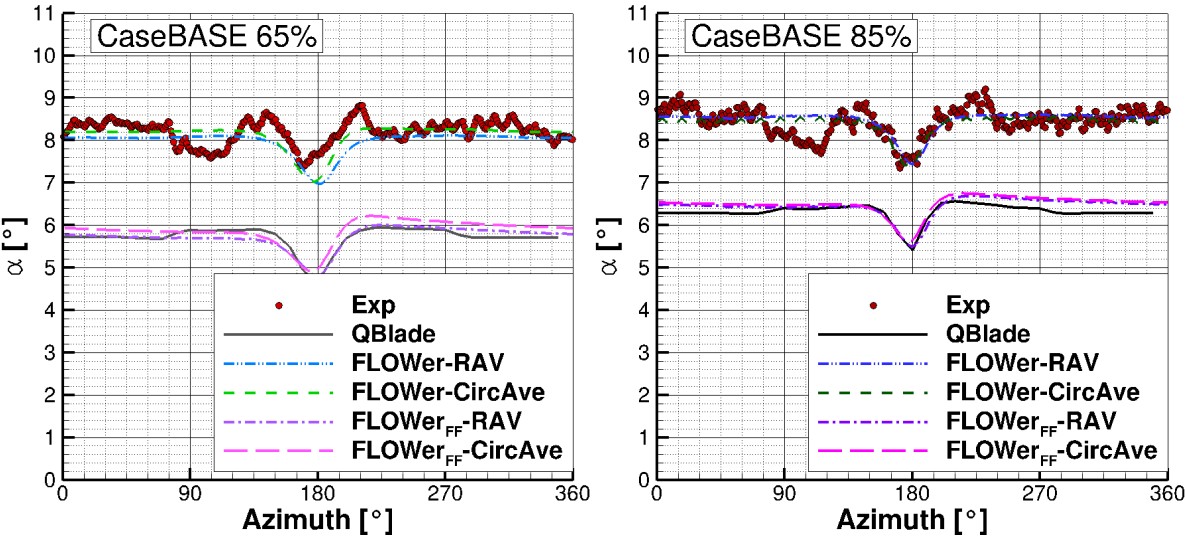

**Figure 15.** AoA distribution over azimuth for $CaseBASE$ for the experiment, *QBlade* and *FLOWer* (*RAV* and *CircAve* for wind tunnel and FF each) at $65\%R$ (left) and $85\%R$ (right).

of the tower is very distinct, due to its relative large diameter, compared to the other components of the turbine. For both, *QBlade* and *FLOWer*, the curve is almost constant before and after this drop. The dip in the experiment at approximately $90°$ azimuth is a result from the traverse, which was located in the test section upstream of the rotor.

Table 10 gives an overview of the differences between experiment and the different simulation results of the averaged angle of attack ($\Delta\overline{\alpha} = \overline{\alpha_{Sim}} - \overline{\alpha_{Exp}}$) for $CaseBASE$ at both probe positions in order to quantify them. In contrast to the on-blade velocity, no relative values were calculated.

There is a good accordance between the experiment and the *FLOWer* results despite the fact that the simulated curves lie

**Table 10.** Differences between the experiment and the different simulation results of the angle of attack for $CaseBASE$.

| $\Delta\overline{\alpha}$ [°] | $QBlade$ | $FLOWer-RAV$ | $FLOWer-CircAve$ | $FLOWer_{FF}-RAV$ | $FLOWer_{FF}-CircAve$ |
|---|---|---|---|---|---|
| $65\%R$ | $-2.48$ | $-0.23$ | $-0.08$ | $-2.48$ | $-2.33$ |
| $85\%R$ | $-2.13$ | $0.03$ | $-0.03$ | $-2.00$ | $-1.95$ |

outside of the measured standard deviation whose average is however small ($\sigma_\alpha(65\%R) = 0.10°$ and $\sigma_\alpha(85\%R) = 0.14°$). Though, they are within the range of the maximum absolute error of $0.8°$, compare subsection 3.2. The larger value for the more outboard region mirrors the effect of the vibrating mounting of the probe. Both AoA evaluation methods for the *FLOWer* solution show almost the same distribution, especially at $85\%$ ($|\Delta\overline{\alpha}_{FLOWer}| \leq 0.23°$ at 65%R and $|\Delta\overline{\alpha}_{FLOWer}| \leq 0.03°$ at 85%R). Reasons for the differences can be attributed to the different approach of the methods (*RAV* is averaging over time and *CircAve* has a local approach, see Jost et al. (2018)). At $65\%$, the level of the AoA is approximately $0.5°$ lower than further outboard for experiment, *QBlade* and *FLOWer*.

An offset of $> 2°$ between the simulation results of *QBlade* and *FLOWer* (including wind tunnel walls) is present for both radial positions. This is a result of the neglection of the wind tunnel walls in the *QBlade* simulation. As the walls impede the expansion of the wake, the velocity in the rotor plane, and consequently the AoA, are higher for the case including wind tunnel. A comparison between the *QBlade* results and the *FLOWer* results under far field condition verifies this assumption, as both the distributions, and the offsets to the measured values, see Table 10, are almost similar. More information about this phenomenon and the underlying reasons can be found in Fischer et al. (2018) and Klein et al. (2018). The small kinks at $\approx 90°$ and $\approx 270°$ azimuth in the *QBlade* results are a result of the usage of the tower model. This model has to be switched on at a certain blade position. In the present simulations this is done as soon as the blade position is located below the nacelle, leading to a discontinuity, which is reduced through interpolation. However, as the tower has a relatively large diameter, the kink can't be completely prevented.

A comparison of the AoA distribution calculated by *QBlade* and *FLOWer* over the normalized radius at azimuth=$0°$ for the wind tunnel and far field cases is shown in Fig. 16.

Again, the influence of the wind tunnel can be seen in the constant offset between the two *FLOWer* cases. As already seen in Fig. 15 and Table 10, the offset between the *RAV* and the *CircAve* results amounts $\approx 0.15°$ at 65%R and decreases to $\approx 0.06°$ at

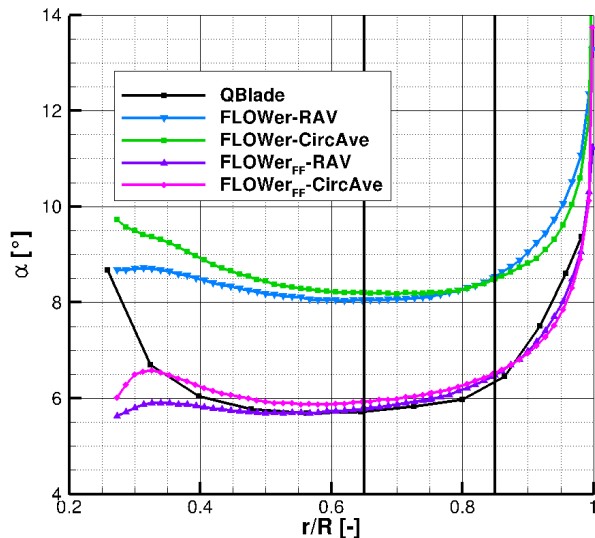

**Figure 16.** AoA distribution over the normalized blade radius at azimuth=0° for *QBlade* and *FLOWer* (*RAV* and *CircAve* for wind tunnel and FF each). Black lines indicate the evaluation positions of Fig 15, Fig 17 and Fig 18.

85%R for both cases (far field and wind tunnel). As already mentioned, the differences are a result of the different approaches of the two methods, see Jost et al. (2018). Between approximately 40% and 90% of the radius, there is a good accordance between the *QBlade* and the *RAV* solution of the *FLOWer* far field case.

Fig. 17 shows the AoA over azimuth under yaw=−15°.

5    The same characteristics as under yaw=0° can also be seen in Fig. 17 under yaw=−15°. Again, the influences of the tower blockage and the traverse are clearly visible. Unlike in $CaseBASE$, the AoA is not constant before and after the drop caused by the tower, due to the yaw misalignment.

In Table 11, an overview of the differences between experiment and the different simulation results of the averaged angle of attack for $CaseYAW15$ at both probe positions is given.

As in $CaseBASE$, the *FLOWer* results show a good agreement to the measurements at both radial positions ($|\Delta\overline{\alpha}_{FLOWer}| \leq$

**Table 11.** Differences between the experiment and the different simulation results of the angle of attack for $CaseYAW15$.

| $\Delta\overline{\alpha}$ [°] | $QBlade$ | $FLOWer - RAV$ | $FLOWer - CircAve$ |
|---|---|---|---|
| $65\%R$ | $-2.05$ | $-0.18$ | $0.01$ |
| $85\%R$ | $-1.76$ | $0.13$ | $0.07$ |

0.18° at 65%R and $|\Delta\overline{\alpha}_{FLOWer}| \leq 0.13°$ at 85%R) and the average of the measured deviation is again small and similar the to

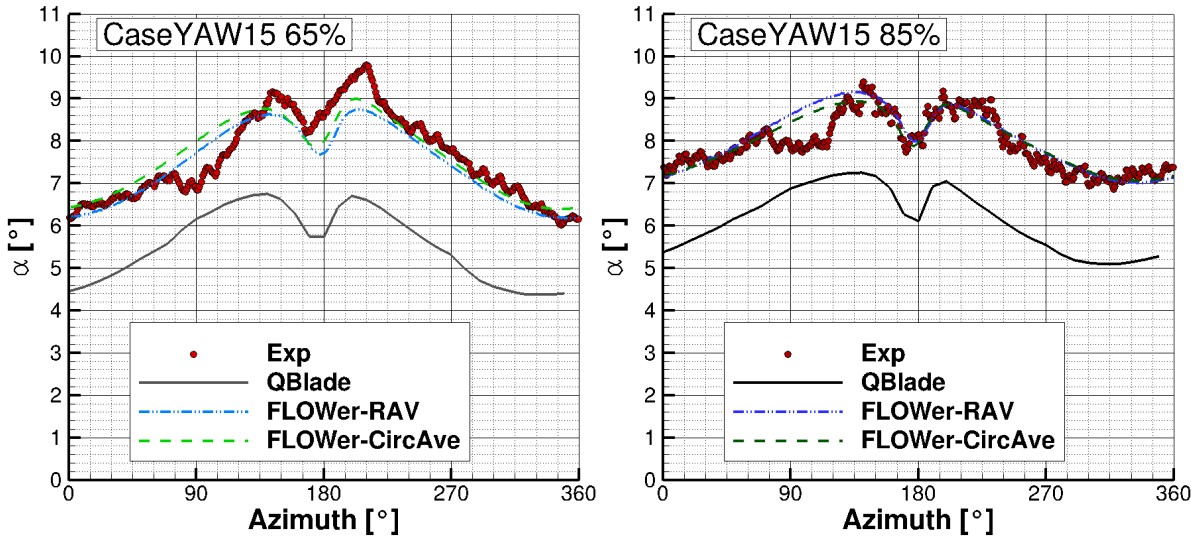

**Figure 17.** AoA distribution over azimuth for $CaseYAW15$ for the experiment, *QBlade* and *FLOWer* (*RAV* and *CircAve*) at $65\%R$ (left) and $85\%R$ (right).

values for the $CaseBASE$ ($\sigma_\alpha(65\%R) = 0.10°$ and $\sigma_\alpha(85\%R) = 0.14°$). Again, the differences of the *CFD* results including wind tunnel are smaller than the maximal absolute error of $0.8°$. The two different evaluation methods for *FLOWer* show almost the same results, too. The difference between the two radial positions amounts approximately $1°$ for all setups. The offset between *QBlade* and *FLOWer* is $> 1.8°$ and but smaller than for case $CaseBASE$ but can still be attributed to the influence

of the wind tunnel walls. The reduction of the difference between *QBlade* and *FLOWer* is a result of the yaw misalignment. Through the rotation of the rotor plane out of the inflow plane, the projected plane gets smaller, leading to a smaller blockage in the wind tunnel. As the change of the projected area follows the cosine-function, the changes in the differences are not linear. As already mentioned, a far field case under yaw misalignment for *FLOWer* was not simulated. The kinks at $\approx 90°$ and $\approx 270°$ azimuth are still present, but less pronounced.

In Fig. 18 the AoA distribution over azimuth for a yaw misalignment of $-30°$ can be seen.

   The effects of the tower blockage and the traverse are still visible. The effects caused by the yaw misalignment are more pronounced here.

An overview of the differences between experiment and the different simulation results of the averaged angle of attack for $CaseYAW30$ at both probe positions is given in Table 12.

At $65\%$, there is a difference between the measurement and *FLOWer* results at the downward moving blade (azimuth=$0°$-$180°$), probably due to the traverse placed in the wind tunnel, whereas there is a good agreement at the upward moving blade (azimuth=$180°$-$360°$). The average accordance between the experiment and the *FLOWer* simulations is satisfactory, as the differences are small ($|\Delta\overline{\alpha}_{FLOWer}| \leq 0.23°$). Further outboard, the curves correspond very well over the whole revolution ($|\Delta\overline{\alpha}_{FLOWer}| \leq 0.12°$), except for the dip at $90°$ azimuth. The average of the deviation amounts $\sigma_\alpha(65\%R) = 0.09°$ and

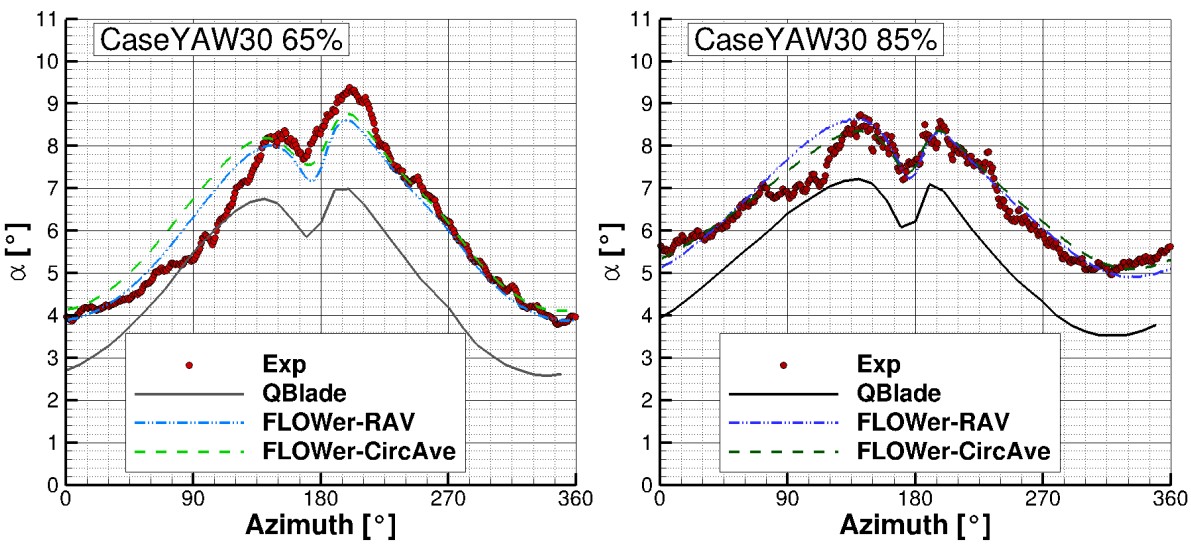

**Figure 18.** AoA distribution over azimuth for $CaseYAW30$ for the experiment, *QBlade* and *FLOWer* (*RAV* and *CircAve*) at $65\%R$ (left) and $85\%R$ (right).

**Table 12.** Differences between the experiment and the different simulation results of the angle of attack for $CaseYAW30$.

| $\Delta\overline{\alpha}\,[°]$ | $QBlade$ | $FLOWer - RAV$ | $FLOWer - CircAve$ |
|---|---|---|---|
| $65\%R$ | $-1.32$ | $-0.02$ | $0.23$ |
| $85\%R$ | $-1.25$ | $0.11$ | $0.12$ |

$\sigma_\alpha(85\%R) = 0.13°$, which can be considered as small. The offset between *QBlade* and *FLOWer*, due to the missing wind tunnel walls in *QBlade*, has decreased and amounts now $< 1.6°$.

For all three cases ($CaseBASE$, $CaseYAW15$ and $CaseYAW30$) at both radial positions, despite the constant offset to the *QBlade* results, the amplitude and phase of the AoA of experiment, *QBlade* and *FLOWer* have a good agreement.

### 4.4 Investigation of the bending moments

In the following, the flapwise bending moments (out-of plane, $M_y$) for one blade, simulated with *QBlade* and *FLOWer*, are compared to each other for all three cases. Fig. 19 shows the curves for $CaseBASE$ (upper left), $CaseYAW15$ (upper right) and $CaseYAW30$ (lower middle).

10 As the forces and moments mainly depend on the AoA, the same characteristics (tower shadow, influence of yaw misalignment,...) like in Fig. 15, Fig. 17 and Fig. 18, can be seen in Fig. 19, as they cascade down from the AoA to the loads.

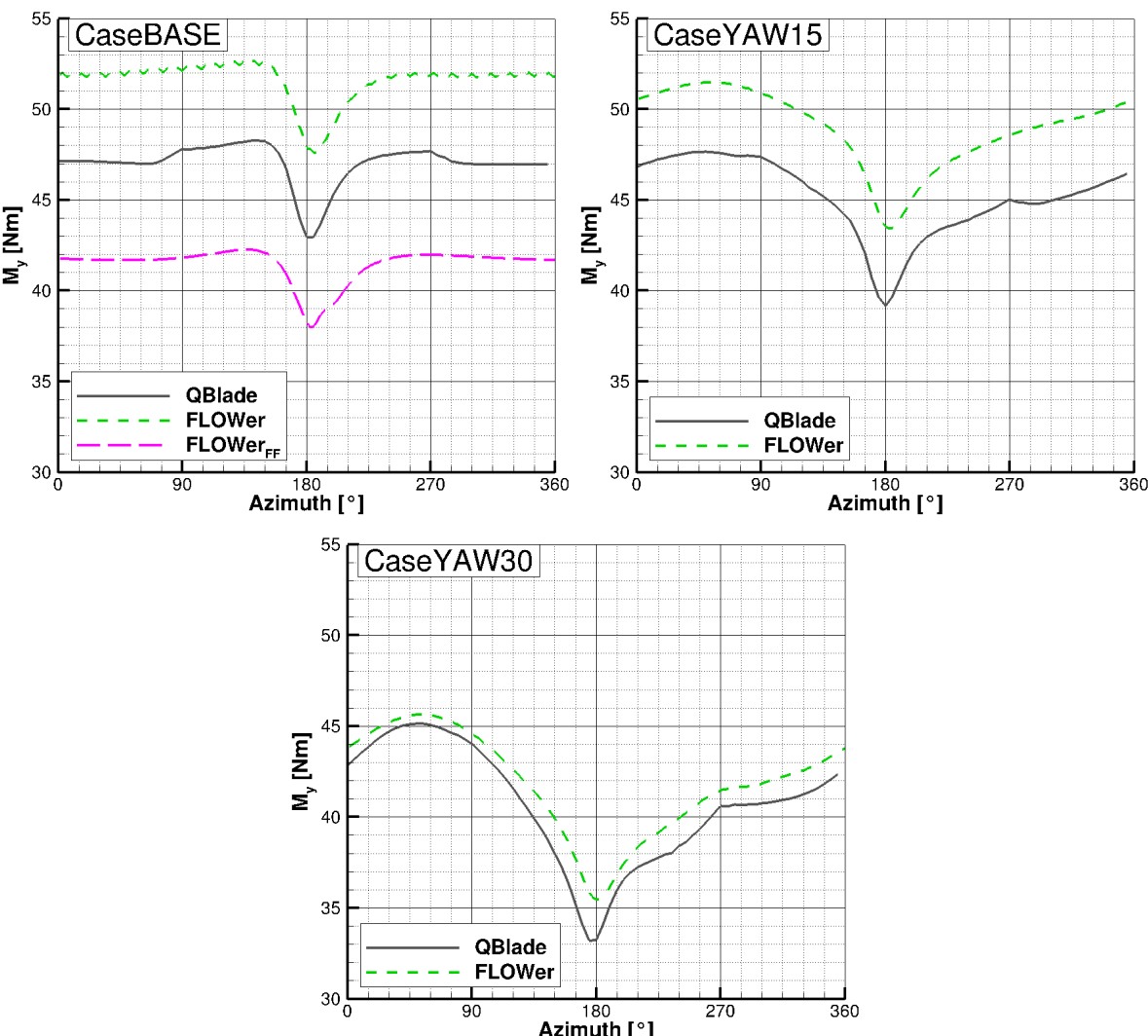

**Figure 19.** Simulated flapwise bending moment ($M_y$) over azimuth for $CaseBASE$ (upper left), $CaseYAW15$ (upper right) and $CaseYAW30$ (lower middle) for *QBlade* and *FLOWer*.

In Table 13, the relative differences between the simulation results of the flapwise bending moment are displayed. The difference between the two *FLOWer* results for the baseline case (upper left figure, $\approx 20\%$) represents the influence of the wind tunnel walls. However, this time, the accordance between the *QBlade* results and the *FLOWer* wind tunnel case ($< 9\%$) is slightly better than between the *QBlade* case and the *FLOWer* far field case. This unexpected result might be a result of the choice of the *XFOIL* polars used for the present *QBlade* simulations, because although the AoA are similar between *QBlade* and $CaseBASE_{FLOWer-FF}$ (see Fig. 15 and Table 10), the bending moments differ. Comparisons of the radial moment distribution and of the force coefficient over the azimuth could lead to a better understanding and assessment of the differences.

**Table 13.** Relative differences between the different simulation results of the averaged flapwise bending moment with respect to the *FLOWer* solution including wind tunnel walls.

| $\Delta \overline{M_y}$ [%] | QBlade | FLOWer$_{FF}$ |
|---|---|---|
| $CaseBASE$ | 8.87 | 19.64 |
| $CaseYAW15$ | 7.86 | — |
| $CaseYAW30$ | 2.81 | — |

The amplitude and phase of the $1p$ frequency, caused by the yaw misalignment, show a good accordance between *QBlade* and *FLOWer* for $CaseBASE$ and $CaseYAW15$. The mean differences under yaw misalignment decrease with increasing yaw angle ($< 8\%$ under $15°$ yaw misalignment and $< 3\%$ under $30°$ yaw misalignment), showing the same tendency as the angle of attack (Table 10, Table 11 and Table 12). Except for the constant offset, the fit between the curves of the *QBlade* and *FLOWer* simulations is similar to the one for the on-blade velocity and the angle of attack. This time, the kinks in the curves at $\approx 90°$ and especially at $\approx 270°$ are a bit more pronounced. For all three cases, *QBlade* predicts, due to the missing wind tunnel walls, smaller values than *FLOWer*.

The comparison of the edgewise bending moments (in plane, $M_x$) can be found in Fig. 20.

The same characteristics of the curves as for the flapwise bending moments (see Fig. 19) can be found in the simulated edgewise bending moments.

The relative differences between the different simulation results for the edgewise bending moments are summarized in Table 14.

The differences between the *FLOWer* results with and without wind tunnel walls are larger than for the flapwise bending

**Table 14.** Relative differences between the different simulation results of the averaged edgewise bending moment with respect to the *FLOWer* solution including wind tunnel walls.

| $\Delta \overline{M_x}$ [%] | QBlade | FLOWer$_{FF}$ |
|---|---|---|
| $CaseBASE$ | 20.82 | 33.37 |
| $CaseYAW15$ | 19.04 | — |
| $CaseYAW30$ | 10.67 | — |

moment ($\Delta \overline{M_x} \approx 33\%$ compared to $\Delta \overline{M_y} < 20\%$, see Table 13). This corresponds to the results of Fischer et al. (2018) and Klein et al. (2018), who also experienced a stronger influence of the walls on the power than on the thrust. The reason for this phenomenon is attributed to the different sensitivity of the forces to AoA variations. The tangential force, which is the main driver of the in plane moment, is more prone to changes in the angle of attack compared to the normal force. Consequently,

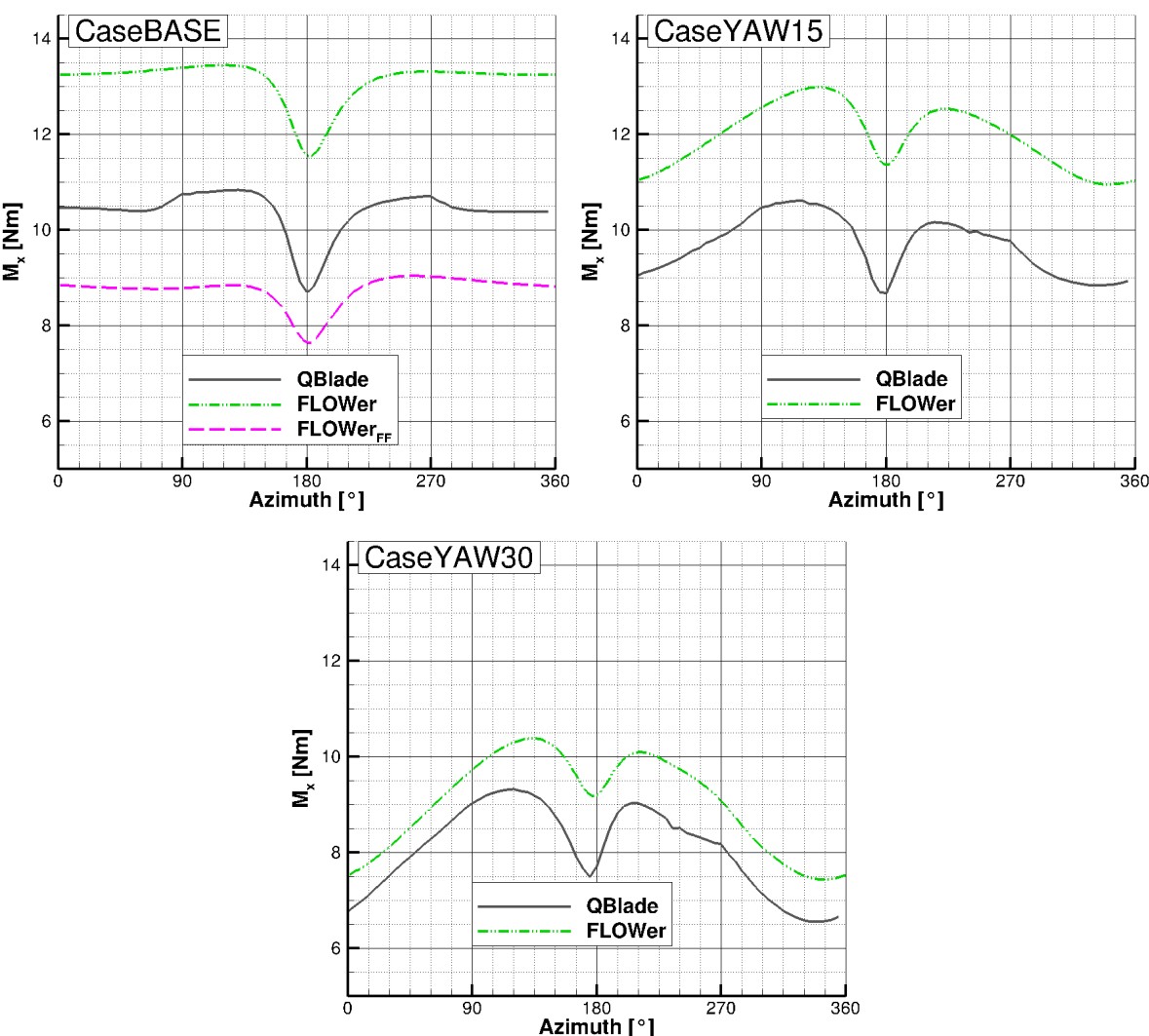

**Figure 20.** Edgewise bending moment ($M_x$) over azimuth for $CaseBASE$ (upper left), $CaseYAW15$ (upper right) and $CaseYAW30$ (lower middle) for experiment, *QBlade* and *FLOWer*.

small differences in the AoA lead to larger deviations in $F_T$ than in $F_N$. Other than for $M_y$, the *QBlade* results for $M_x$ are closer to the *FLOWer* far field results than to the wind tunnel results. The progression of the edgewise bending moment is almost similar between *QBlade* and *FLOWer* for all three inflow directions. The mean differences under $15°$ yaw misalignment ($\approx 19\%$) are slightly smaller than for $CaseBASE$ ($\approx 21\%$), but the difference under $30°$ yaw misalignment is significantly smaller ($< 11\%$) than for the other two cases. Again, the change in the projected area and the blockage in the wind tunnel can be alluded as reason for this tendency.

To sum up, the progression of the curves fit quite good for both moments, except the kinks caused by the tower shadow model

in *QBlade*. The offset between the results seem to depend on to consideration of the wind tunnel walls and the chosen polar set in *QBlade*. The decreasing differences between *QBlade* and *FLOWer* with increasing yaw misalignment is a result of the decreasing projected rotor plane which influences the blockage in the wind tunnel.

## 5  Summary

Experimental und numerical investigations of a model wind turbine, placed in a wind tunnel with high blockage ratio, were presented in the present paper. Thereby, two codes of different fidelity were used. In the simulations conducted with the Lifting Line Free Vortex Wake code *QBlade*, the wind tunnel walls had to be neglected and the turbine was simulated under far field condition. Unsteady Reynolds-averaged Navier-Stokes simulations have been performed with the Computational Fluid Dynamics code *FLOWer*. Thereby, a far field case, as well as simulations including the wind tunnel walls, were investigated. In all simulations, the tower was considered, but they have been performed under uniform inflow, neglecting the turbulent inflow in the experiment.

The experiments provided validation data and the comparison between experiment and the *FLOWer* wind tunnel case aimed at the validation of the *CFD* simulation. Through the comparison between two *FLOWer* cases (wind tunnel and far field) the influence of the blockage ratio was assessed. With the knowledge about the influence of the wind tunnel walls, the suitability of the *LLFVW* code to perform preliminary investigations for future studies with the model wind turbine could be investigated by a the comparison between *QBlade* and the *FLOWer* far field case.

A comparison between the measured flow fields and the velocity planes extracted from *FLOWer* simulations including wind tunnel walls was conducted. Thereby, two different velocity planes were investigated. One is located $0.43d$ upstream of the turbine, one $0.5d$ downstream. The velocity fields upstream of the turbine showed a good agreement in the rotor area, as the average deviation amounts about $3\%$ of the inflow velocity. Downstream of the rotor plane, the differences were more pronounced (mean deviation $\approx 7\%$ of the inflow velocity). The areas of the tip vortices and the wake of the nacelle are most prominent. The differences between the experimental and numerical results upstream and downstream are caused, amongst other, by vertical shear and higher turbulence in the measurements. Additionally, the differences in the wake of the nacelle and the outer region of the rotor might be caused by the high flow angles influencing the hot wire measurement downstream of the rotor.

At two radial positions ($65\%$R and $85\%$R), the on-blade velocity and the AoA were measured with 3-hole probes and compared to the results obtained from *QBlade* and both *FLOWer* cases. For the investigation of these parameters, three different yaw cases (yaw=$0°$; $-15°$ and $-30°$) were considered.

The mean deviations of the on-blade velocity between the experiment and each simulation are $< 4\%$ at $65\%$ of the radius and $< 2\%$ at $85\%$ of the radius.

The AoA calculated with *FLOWer* including wind tunnel showed a good agreement with the experimental results, as the maximum mean difference amounts $0.23°$. As the *QBlade* results and the *FLOWer* simulation without wind tunnel walls are almost similar, the constant offset of approximately $1°$-$2°$ between the experiment and the far field simulations is a result of the ne-

glection of the wind tunnel walls.

Finally, the blade root bending moments are compared between *QBlade* and the two *FLOWer* cases. For the out-of plane bending moment, the difference between the two *FLOWer* cases (far field and wind tunnel) can be accredited to the influence of the wind tunnel walls. The offset between the *QBlade* results and both *FLOWer* cases can not only be attributed to the influence of

the wind tunnel walls. As the bending moments differ between the two far field cases despite the good accordance concerning the AoA, the chosen set of airfoil polars, which is used in the *QBlade* simulations, influences the loads. The accordance between the calculated amplitude and phase of *QBlade* and *FLOWer* is good.

The same conclusions as for the flapwise bending moment can be drawn for the edgewise bending moment. However, the relative deviations between the simulated curves of *QBlade* and *FLOWer* are larger.

To sum up, a good accordance was achieved for the absolute values and the azimuthal distribution regarding the on-blade velocity and the AoA. Consequently the numerical setup of *FLOWer* can be seen as validated in terms of these two parameters. Concerning the velocity planes, differences between experiment and *FLOWer* occur but can be explained. The comparison between the two *FLOWer* cases (with and without wind tunnel walls) showed, that in the present case, the wind tunnel leads to a constant offset between the curves for the on-blade velocity, the AoA and the bending moments. Regarding the *QBlade*

results, the on-blade velocity, as well as the amplitude and phase of the AoA can be seen as validated by the experiment, too. As the AoA distribution of *QBlade* lies on the far field solutions of *FLOWer*, the differences in the mean values of the AoA can be attributed to the absence of wind tunnel walls in the *QBlade* predictions. The offset between *QBlade* and *FLOWer* wind tunnel case regarding the bending moments is not only a result of the neglection of the walls, but is also influenced by the set of airfoil polars used in the *LLFVW* simulation.

In a next step, in order to better match the experimental conditions, simulations with unsteady inflow, considering the measured shear and turbulence, will be performed. Moreover, experiments with passive and active load control will be performed and compared to simulations of both, *QBlade* and *FLOWer*. Thereby, *QBlade* will be used for dimensioning purposes of the flaps prior to the experiments. Afterwards, the most promising configurations will be investigated numerically on a full size turbine by *QBlade* and *FLOWer*, where the *LLFVW* code can be used for the preliminary design, and the *CFD* code for the closer look

into the aerodynamic details.

*Data availability.* Measurement data and simulation results can be provided by contacting the corresponding author or Thorsten Lutz (lutz@iag.uni-stuttgart.de).

*Competing interests.* The authors declare that they have no conflict of interest.

*Acknowledgements.* All computational resources used for the *FLOWer* simulations were provided by the High Performance Computing Center Stuttgart (*HLRS*). The studies presented in this article have been funded by the *German Research Foundation* (*DFG*) and were performed in the course of the *DFG* PAK 780 project. The authors want to thank the reviewers and the editor for their useful suggestions.

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
