# Peer review of "About the suitability of different numerical methods to reproduce model wind turbine measurements in a wind tunnel with high blockage ratio"

_Wind Energy Science, 2017_

## Referee Comment (RC1) · Anonymous Referee #1 · 8 Nov 2017

The paper 'About the suitability of different numerical methods to reproduce model wind turbine measurements in a wind tunnel with high blockage ratio' is motivated by three main objectives: the provision of experimental data for validation of numerical wind turbine simulations, the validation of two specific numerical codes, and the assessment of the wind tunnel blockage effect. These three objectives have a significant scientific importance for the development and validation of reliable numerical codes and for wind tunnel studies with a high blockage ratio. The work in this paper is part of a larger wind energy research project about active flow control for wind turbines. Several papers have already been published as part of this project. The added value of this specific paper is the presentation of experimental data for the on-blade velocity, angle-of-attack and blade bending moment, as a function of azimuth, and the comparison with two numerical codes: a lifting line free vortex wake code called QBlade and a URANS flowsolver called FLOWer. The measurements and comparisons are done for three turbine yaw angles, and the simulations with FLOWer are done with and without wind tunnel walls.

Although the paper presents valuable experimental data and a comparison with two numerical codes, it requires in my opinion a major revision before it can be considered for publication. In general, I find the text not always clear, unstructured, and in many cases too vague. The lack of structure makes the text confusing to read, especially because the paper is investigation many different parameters, such as experiments, two different numerical codes, blockage effects and the effect of yaw misalignment. The new content of this paper is presented in chapter 3, with a comparison between experiments and simulations. Although the results are very interesting, the discussion is not thorough enough and the conclusions oversimplified. The differences between the experimental and numerical inflow conditions are neglected. However, the inflow conditions show a difference in the mean velocity of +- 10% compared to the simulation conditions, which seems not negligible. The authors should discuss the consequences of the different inflow conditions on the measurements in more detail. Furthermore, it would be interesting to verify and discuss specifically which physics are modeled correctly by the codes, and which not. For example, one could verify this by estimating the angle of attack, on-blade velocity, and bending moment with a simple BEM method, to verify the benefits of the Lifting Line Free Vortex Wake code. The comparisons for the angle-of-attack and on-blade velocity show a good agreement between simulations and experiments. However, the comparison for the bending moment, which is a result of the former two parameters, shows a surprisingly large difference. It would be interesting to discuss the possible causes for this difference in more detail. I have added a list of major and minor comments below.

Major comments:

- The introduction does not clearly motivate the research objectives with a literature overview. For instance on page 2, lines 13-15, it is mentioned that earlier studies already verified the effect of wind tunnel walls and blockage effects with simulations. What where the conclusions? From the introduction it is thus unclear what this paper will contribute to the study of wind tunnel blockage effects.
- When a comparison with measurements is done, it is important to consider the measurement uncertainty. Add an estimation of the measurement uncertainty.
- Section 2 about the 'methodology and setups' is badly organized and needs a significant improvement.
  - Each section and sub-section needs an introduction.

- The section about the wind tunnel is too short and brief.
- It is not necessary to mention the top-speed for the wind tunnel test section which is not used in this paper.
- The measurements are performed in the settling chamber which has as purpose to condition the non-homogeneous and turbulent flow from the wind tunnel fan before it enters the test section. As figure 9 indicates, there is a significant mean shear over the cross section in the settling chamber, and the turbulence intensity is not negligible. It is important to provide a motivation for this configuration, provide a characterization of the inflow and turbulence properties, and discuss the effects it may have on the results.
- Section 2.1 'Experimental setup' does not describe the velocity measurement setup.
- Mention the specific acquisition devices, and not just the name of the manufacturer.
- P5 Figure 3: The actuators for the flaps and the 3-hole probes + air tubes on the smart blade look like they will influence the airflow around the blade. The presence and impact of this blockage should be discussed.
- P10 L3: An acquisition time of 16 seconds is short for velocity measurements. Please motivate, e.g. based on the integral time scale, that this is sufficiently long for good statistics.
- P15 L5: I don't agree with the statement that the error is small. The error in figure 12 is higher than 10% in a large part of the wake: shear region and center.
- Figure 18: The experiments show a significant dip around 90 degrees. This is not visible in the simulations. Is there a reason for this effect? Is this also due to the traverse? Explain the situation.
- The differences between the measured and simulated bending moments in figures 20 and 21 are significant. It is not ok to say that this is a good agreement. The experimental curves follow a different pattern, especially for CaseBase. Is there an explanation for this?
- The authors should be careful with copyrights. For instance figure 1a, figure 2, figure 3b and figure 8 can be found identically in the paper 'Reproducible inflow modifications for a wind tunnel mounted research HAWT' by Bartholomay S., et al. 2017.
- P24 L26: The main conclusion of this paper is too strong. The experiments have too many differences (e.g. vertical shear and turbulence) to make this statement. Furthermore, the agreement for the bending moment is not good at all. Instead make conclusions on what can be estimated correctly, what not, and which physics are modeled correctly.

Minor comments:

- Define abbreviations at first use in the main text. Don't define abbreviations in the abstract, and limit the use of abbreviations in the abstract. For example CFD, LLFVW, URANS, ..
- Throughout the text, several sentences are unnecessarily long, or have a structure where the subject is placed at the end, which can be confusing. Improving these sentences will benefit the clarity of the text. Some examples are:
  - P1, L1
  - P1, L2: This is a long sentence and not entirely clear. For instance 'methods of different fidelity' is too vague.

- P1, L5: Is it relevant where the code was run?
- P2, L3: long sentence.
- P2 L11: It is not clear what 'a one third model' is + this is a long sentence.
- P2 L17: The yaw angle is negative. Does this matter? The orientation is not mentioned.
- P2 L19: 'the flow around the rotor' is too vague.
- Define the term 'far field conditions'.
- Units need to be formatted correctly.
- The first sentence of the introduction is too vague and unnecessary.
- P1 L17: You mention 'simulations' but the referenced paper presents experimental results.
- P2 L16: Instead of 'three different states', it would be more clear to mention 'Three different yaw-misalignment cases'
- P2 L23: This sentence is very long, consider breaking it up in several more clear and well defined sentences.
- P3 L6: The text mentions 'low Reynolds numbers'. Please describe the Reynolds numbers at which the experiments are run, and motivate if the experiments scale realistically.
- P3 L9: How is the boundary layer thickness estimated? Is it possible to indicate the tape on figure 3?
- P3 L15: It is confusing to mention at this point in the text the overall goal of the research project, as it is different from the objectives of this paper.
- P5 L2: What is meant with 'trailing edge deployment'?
- Table 2: What are the units for the wake length?
- P6 L14: 21 panels are mentioned in the text, but in Table 2 15 panels are mentioned. Which one is correct?
- P7 l3: It is not entirely clear what is meant with 'overlapped using the CHIMERA technique', overlapping several grids?
  - P7 L7: Don't use double brackets '(  ()  )'.
- Table 3: I suppose the units are millions of cells?
- P 8 L6-7: This is a repetition.
- Table 4 is an important table. Maybe it can be discussed earlier in the text.
- P10 L3: Which probe was taken as the reference then? How are the hot-wire probes calibrated?
- Figure 7 is unnecessary.
- P11 L15: How much are these corrections typically? Maybe indicate in figure 16.
- P12: The description of the strain gage setups should be done in the experimental setup section.
- P9 L17 The text mentions measurements at 1.05d , while P13 L3 doesn't mention measurements at 1.05D. Be consistent, also with the unit of 'D'.
- P13 L15 'More information about this topic can be found in..' is too vague.
- P14 L6: 'Some aspects' is too vague.
- P14 L10-15: Conclusions on wake comparison are not clear, which of the two simulations is discussed?
- P15 L12: What does this mean for the measurement blade: isn't 100% the maximal radial position?
- P17 L8-10 & p18 L10: 'More information about..' is too vague.
- P18 L13-L16: This should be mentioned in the introduction.

---

## Referee Comment (RC2) · Anonymous Referee #2 · 3 Jan 2018

Review of Manuscript # wes-2017-42

Title: About the suitability of different numerical methods to reproduce model wind turbine measurements in a wind tunnel with high blockage ratio by A. C. Klein et al.

In the manuscript, the authors performed numerical and experimental studies of a model wind turbine in the wind tunnel exposed to high blockage ratio. Two different numerical frameworks are considered, and the obtained results are compared with the experimental data. Unfortunately, the paper suffers from major issues that require a major revision before the paper can be considered for the publication. The major comments are:

1. The introduction is incomplete. The authors only provided a review of the previous studies done by themselves and their colleagues, and they neglected the key papers and contributions done by the other researchers who worked extensively on this topic (e.g., Research group at JHU, EPFL, KU-Leuven, …).

2. The objective of the work is performing numerical simulations to validate the experimental data. However, the results presented in the paper cannot be considered as a validation. There is a huge discrepancy between the experimental data and numerical results and the authors did not explain the reasons behind that. The authors should perform systematic experimental and numerical experiments with providing a clear explanation of the observed discrepancies. Similar studies have been extensively performed by the other groups which some of them mentioned above. For example, the results presented in Figure 21 cannot be considered as validation. There is a huge difference between the experimental data and simulation results. Besides, the authors mentioned: "The curve for the baseline blade is missing in the current plot as the sensors had a malfunction during the measurement." This statement is not acceptable for a paper that is going to be published in a journal. The same trend is also presented in the other figures. No clear explanations are provided about the differences.

3. The authors mentioned: "As it is currently not possible to include the wind tunnel walls into the LLFVW simulations of QBlade, far-field simulations were conducted." Since the objective of the paper is exploring the blockage effect, it is not clear what the purpose of having the results from the QBlade is. It would be more relevant if the QBlade results considering the wind tunnel wall are added to the paper. Otherwise, it is not needed to add the results from another code that is not consistent with the experimental investigation.

4. As it is mentioned before, the validation section is not acceptable. Also, since Figures 9 and 10 are qualitative results, the authors need to provide more quantitative comparisons by, for instance, comparing the results at different locations. Although, even from the contour plots, the agreement is not good. Also, besides the mean velocity, the variances obtained from both the experiment and numerics should be provided. This is a very standard way for validation of numerical tools against the experimental data.

5. Figure 13-16, it seems that the y-axis chosen here is too wide to minimize the difference between the experiments and numerics. For example, in Fig. 13 (left), it is trivial that the on-blade velocity cannot be ranged from 0 to 30. In particular, considering the tip-speed ratio and the incoming wind speed, it should be in a much narrower range.

6. Most of the citation about the numerical frameworks are technical report, conference proceeding or personal communications. Typically, it is expected that the papers cited in the manuscript were peer-reviewed before.

7. The incoming flow is not characterized in the manuscript. The information about the incoming wind, the associated turbulence level, the Reynolds number based on the chord length is missing in the manuscript.

8. The results are provided without any sensitivity analysis to the grid resolutions. As mentioned before, the agreement between the numerical results and experiment is poor. Although the code might have tested before for other cases, it is required to perform the grid resolution sensitivity for this particular analysis presented in the manuscript. The convergence of the statistics also should be provided.

---

## Author Comment (AC1) · 31 Mar 2018

Dear Reviewer, please find attached two documents, which contain the author comments. Kind regards, Annette Klein

Please also note the supplement to this comment:
https://www.wind-energ-sci-discuss.net/wes-2017-42/wes-2017-42-AC1-supplement.zip

---

## Author Response (AR1)

**This document includes:**

1. Point-to-point response to the first reviewer

2. Point-to-point response to the second reviewer

3. List of the major changes in the manuscript

4. Marked-up manuscript
   - changed sections with regard to the comments by reviewer 1 are marked in yellow
   - changed sections with regard to the comments by reviewer 2 are marked in orange
   - changed sections with regard to comments by both reviewers are marked in gray
   - changes with regard to no comments but which serve a better understanding and an improvement of the manuscript are marked in green

**Reply to the comments of Reviewer No. 1**

Annette Claudia Klein on behalf of the authors
IAG, University of Stuttgart

March 31, 2018

The authors would like to thank the reviewer for his/her efforts and constructive comments. They are very much appreciated and incorporated into the revised manuscript.

In this document the comments given by the 1st reviewer are addressed consecutively. The following formatting is chosen:

- The reviewer comments are marked in blue and italic.

- The reply by the authors is in black color.

- A marked-up manuscript is added. Changed sections with regard to the comments by reviewer 1 are marked in yellow. Changed sections with regard to comments by both reviewers are marked in gray. Changes with regard to no comments but which serve a better understanding and an improvement of the submission are marked in green.

Some manuscripts, which were accepted during the review of this manuscript, are now published:
Fischer et al. 2016 is now referred under Fischer et al. 2018
Wendler et al. (2016) is now published.
Klein et al. (2017) is now referred under Klein et al. (2018)

Moreover, Jost (2017) and Klein (2017) are now replaced by Jost et al. (2018)
The display of the references were adopted in the reference list.

We would like to mention, that since the first submission of this manuscript in September 2017, two conference papers, which partly use the same data as in the present manuscript, were written, submitted and accepted for the AIAA 2018 Conference Series. As these papers reference on the present submission, they were not cited here.

Bartholomay, Sirko, et al. "Towards Active Flow Control on a Research Scale Wind Turbine Using PID controlled Trailing Edge Flaps." 2018 Wind Energy Symposium. 2018.

Marten, David, et al. "Numerical and Experimental Investigation of Trailing Edge Flap Performance on a Model Wind Turbine." 2018 Wind Energy Symposium. 2018.

**General comments "G"**

1. "*In general, I find the text not always clear, unstructured, and in many cases too vague. The lack of structure makes the text confusing to read, especially because the paper is investigation*

*many different parameters, such as experiments, two different numerical codes, blockage effects and the effect of yaw misalignment."*

The authors apologize for the inconvenience. In addition to the major and minor comments, the manuscript was completely revised and the authors improved the structure by putting subsection 2.5 into an extra section (Section 3), see R1:G1 (page 13, line 280).

2. *"Although the results are very interesting, the discussion is not thorough enough and the conclusions oversimplified."*

The discussion of the results (section 4) was detailed and more specific conclusions were drawn (section 5) in the revised manuscript. More detailed references can be found in the corresponding comments, e.g. Ma3.4, Ma4 or Ma10.

3. *"The differences between the experimental and numerical inflow conditions are neglected. However, the inflow conditions show a difference in the mean velocity of +- 10% compared to the simulation conditions, which seems not negligible. The authors should discuss the consequences of the different inflow conditions on the measurements in more detail."*

Due to the wind tunnel, the inflow in the experiment is less uniform compared to the simulations and a difference of $-10\%$ occurs in a small region, compare R1:G3 (page 18, line 396). But as the average inflow velocity is the same within the rotor area, the differences in the inflow are lower than $\pm10\%$. Downstream of the rotor, the differences are bigger, especially in the wake of the nacelle and in the area of the tip vortices, compare Fig. 11. According to Major Comment Ma3.4 and Ma6, and to Major Comment Ma4 and Ma7 of reviewer 2, additional information were added and the consequences were discussed, for example in Table 5 and in Table 6.

4. *"Furthermore, it would be interesting to verify and discuss specifically which physics are modelled correctly by the codes, and which not. For example, one could verify this by estimating the angle of attack, on-blade velocity, and bending moment with a simple BEM method, to verify the benefits of the Lifting Line Free Vortex Wake code."*

The authors didn't estimate the angle of attack, on-blade velocity, and bending moment with a simple BEM method but prefer to describe the advantages of the vortex codes over traditional BEM methods in the revised manuscript. Moreover, references about the advantages, especially in unsteady operating conditions, were added, compare R1:G4 (page 9, line 200).
The modelled physics are mentioned with regard to Major Comment Ma10.

5. *"However, the comparison for the bending moment, which is a result of the former two parameters, shows a surprisingly large difference. It would be interesting to discuss the possible causes for this difference in more detail."*

The authors agree that a discussion of possible causes would be interesting. However, according to the suggestion of reviewer 2 (Ma2), the measured bending moments were removed from the present manuscript because of the large fluctuations and the need of strong filtering, which influenced the bending moments considerably. See also Major Comment Ma8.

**Major comments "Ma"**

1. *"The introduction does not clearly motivate the research objectives with a literature overview. For instance on page 2, lines 13-15, it is mentioned that earlier studies already verified the effect of wind tunnel walls and blockage effects with simulations. What where the conclusions? From the introduction it is thus unclear what this paper will contribute to the study of wind tunnel blockage effects."*

The introduction was completely revised.

The objective concerning the influence of the walls on the results was reformulated in the abstract, see $\boxed{\textbf{R1:Ma1-b}}$ (page 1, line 3) and in the introduction, compare $\boxed{\textbf{R1:Ma1-e}}$ (page 3, line 65)). As the reviewer noted, the effect of the wind tunnel walls were already verified and estimated in other papers (Fischer et al. (2018), Klein et al. (2017)). The conclusions of these papers were now added in the present manuscript, see $\boxed{\textbf{R1:Ma1-a}}$ (page 3, line 58). However, the published papers concerning the blockage effect present pure numerical results. No comparisons with experimental data to validate the simulated effect were done, see $\boxed{\textbf{R1:Ma1-c}}$ (page 3, line 62). So in the present submission, the gain in knowledge is on the one hand the estimation of the influence assessed with FLOWer, see $\boxed{\textbf{R1:Ma1-e}}$ (page 3, line 65). On the other hand, in QBlade, the walls were not taken into account. Therefore, after the validation of the influence of the wind tunnel walls in FLOWer, these simulations can be used for a comparison with FLOWer far field simulations to estimate the effect of the walls, see $\boxed{\textbf{R1:Ma1-d}}$ (page 33, line 618). Afterwards, a comparison with the QBlade results can be done. Without the link with FLOWer, a comparison of the experimental results and the simulated QBlade results would not be that meaningful. But with the influence of the wind tunnel estimated, the different results can be compared and interpreted.

Moreover, the literature overview was extended. As suggested by reviewer 2, articles from research groups from JHU, EPFL and KU-Leuven have been taken into account. References about hot-wire measurement to investigate the wake under different operating conditions like yaw misalignment can be found at $\boxed{\textbf{R1:Ma1-j}}$ (page 1, line 22). Further information about the measurement of mean velocity and turbulence intensity was integrated at $\boxed{\textbf{R1:Ma1-f}}$ (page 2, line 25). References about further applications and benefits of wind tunnels can now be found at $\boxed{\textbf{R1:Ma1-g}}$ (page 2, line 31). References about the investigation of the blockage ratio were added at $\boxed{\textbf{R1:Ma1-h}}$ (page 2, line 43) and $\boxed{\textbf{R1:Ma1-i}}$ (page 2, line 49).

2. "*When a comparison with measurements is done, it is important to consider the measurement uncertainty. Add an estimation of the measurement uncertainty.*"

The velocity planes are measured with hot-wire probes. These probes were calibrated using a KIMO L-type pitot tube with an error of 1% corresponding to 0.1 m/s at the maximum calibrated velocity. The dynamic pressure was measured using a Baratron (pressure sensor) with an error of 0.15% of the pressure reading. The latter corresponds to a velocity error of 0.07% = 0.0075 m/s at the maximum calibrated velocity. Therefore, the error due to the calibration error is 0.1075 m/s. In order to assess the bias error between the probes, in all measurements of the campaign, a set of 20 measurement positions was measured by each probe. Thereby, the bias between the probes was determined. The maximum bias of the probes was determined from the entire dataset as 0.33 m/s, corresponding to 3.3% in reference to maximum calibrated velocity of 10m/s. This is in good agreement to the error estimation shown in the Dantec User Guide, Finn (2002), where the velocity error is calculated to 3%. Summarizing, the error due to calibration and the hot-wire measurement chain is 4.37%, corresponding to 0.44 m/s.

This information is now added in the manuscript, see $\boxed{\textbf{R1:Ma2-b}}$ (page 14, line 297).

Moreover, the simulated and measured averaged standard deviation for the velocity planes can now be found in Table 5 and Table 6 and values are now mentioned and discussed in the text, see $\boxed{\textbf{R1:Ma2-i}}$ (page 17, line 387) and $\boxed{\textbf{R1:Ma2-j}}$ (page 19, line 423).

The on-blade velocity and angle of attack are measured by three-hole probes on the rotor blade. In order to assess the error of these measurements, the measurement chain was analyzed. The three-hole probes were separately (detached from the rotor blade) calibrated in a calibration-setup. The relative error for the velocity measurement amounts 1% in reference to the maximum speed of 22 m/s, which corresponds to 0.22 m/s. In a range of -30deg to +30deg, the maximum error (bias) for the AoA is 1.6%/Fullscale, which corresponds to 0.48deg for the considered velocity range. To assess the error of the induction correction, the probe was installed in a 2d-wing setup, which ist mounted on a turn table. Thereby, the velocity was compared also to the inflow velocity, that was measured by a differential pressure measurement along the duct section upstream of the test-section. An error of 2%-2.7% is present in the linear region, corresponding to an error of 0.3 to 0.4 m/s. The calculated AoA are compared to the AoA that is set by the turn-table. Within the linear attached flow region, the error of the AoA measurement remains below 2% in reference to the 40deg pitched airfoil, which corresponds to 0.8deg. As the error for the induction correction includes the error which is created by a 'pure' three-hole probe setup, the maximal errors of the velocity and AoA measurement can be summarized for the linear attached flow region as $\Delta AoA = 0.8deg$ and $\Delta v = 0.4m/s$.

This information is now added in the manuscript, see $\boxed{\textbf{R1:Ma2-a}}$ (page 15, line 329).

Moreover, information about the average standard deviation of the measured on-blade velocity is now mentioned in the text, see $\boxed{\textbf{R1:Ma2-c}}$ (page 21, line 451), $\boxed{\textbf{R1:Ma2-d}}$ (page 22, line 476) and $\boxed{\textbf{R1:Ma2-e}}$ (page 24, line 493). The averaged standard deviation for the measured AoA was added as well, see $\boxed{\textbf{R1:Ma2-f}}$ (page 25, line 512), $\boxed{\textbf{R1:Ma2-g}}$ (page 26, line 542) and $\boxed{\textbf{R1:Ma2-h}}$ (page 29, line 562).

3. "*Section 2 about the 'methodology and setups' is badly organized and needs a significant improvement.*"

The manuscript was reorganized to improve the structure. About the methodology and setups, the general information about the setups were now placed at the beginning, see $\boxed{\textbf{R1:Ma3-a}}$ (page 4, line 90). Afterwards, the different setups are described, starting with the experimental setup. Thereby, the order was from big (wind tunnel) over turbine to small (blade). The description of the two numerical codes starts with information about the code, which is in each case followed by the description of the numerical setup used for the present submission. Afterwards, the data acquisition is introduced with Fig. 6, see $\boxed{\textbf{R1:Ma3-b}}$ (page 13, line 281).

3.1 "*Each section and sub-section needs an introduction.*"

An overall introduction for the whole section as well as short descriptions of the content of each subsection were added, see $\boxed{\textbf{R1:Ma3.1-a}}$ (page 3, line 86), $\boxed{\textbf{R1:Ma3.1-b}}$ (page 4, line 90), $\boxed{\textbf{R1:Ma3.1-c}}$ (page 5, line 106), $\boxed{\textbf{R1:Ma3.1-d}}$ (page 8, line 176), $\boxed{\textbf{R1:Ma3.1-e}}$ (page 10, line 223) and $\boxed{\textbf{R1:Ma3.1-f}}$ (page 13, line 280).

3.2 "*The section about the wind tunnel is too short and brief.*"

The section about the wind tunnel is now more detailed. The authors added more information about the wind tunnel and the composition of the test section which could be helpful to understand the present setup, see $\boxed{\textbf{R1:Ma3.2-a}}$ (page 5, line 111) and $\boxed{\textbf{R1:Ma3.2-b}}$ (page 5, line 115). Further information about the inflow was added in the course of the consideration of Ma3.4.

3.3 "*It is not necessary to mention the top-speed for the wind tunnel test section which is not used in this paper.*"

The information was removed, see $\boxed{\textbf{R1:Ma3.3-a}}$ (page 5, line 111) and $\boxed{\textbf{R1:Ma3.3-b}}$ (page 5, line 116).

3.4 "*The measurements are performed in the settling chamber which has as purpose to condition the non-homogeneous and turbulent flow from the wind tunnel fan before it enters the test section. As figure 9 indicates, there is a significant mean shear over the cross section in the settling chamber, and the turbulence intensity is not negligible. It is important to provide*

*a motivation for this configuration, provide a characterization of the inflow and turbulence properties, and discuss the effects it may have on the results."*

A motivation for this configuration was added, see $\boxed{\textbf{R1:Ma3.4-a}}$ (page 5, line 111). Information about the velocity plane and the occurring inequality were investigated by Bartholomay et al., 2017, and were addressed in Subsection 4.1, see $\boxed{\textbf{R1:Ma3.4-c}}$ (page 16, line 373). More information about the distribution of the turbulence intensity were added, see $\boxed{\textbf{R1:Ma3.4-b}}$ (page 5, line 118).

Additionally, the authors included tables with the streamwise mean velocity, the standard deviation of the streamwise velocity as well as the global turbulence intensity in x-y-direction for both locations, see Table 5 $\boxed{\textbf{R1:Ma3.4-f}}$ (page 17, line 383) and Table 6 $\boxed{\textbf{R1:Ma3.4-g}}$ (page 19, line 418). Moreover, the mean differences between measurement and simulation were determined and mentioned in the text, see $\boxed{\textbf{R1:Ma3.4-h}}$ (page 18, line 395) and $\boxed{\textbf{R1:Ma3.4-i}}$ (page 20, line 430).

An overview on the influence of the turbulent inflow on the results was added, see $\boxed{\textbf{R1:Ma3.4-d}}$ (page 5, line 122) and was already given in Section 4, see $\boxed{\textbf{R1:Ma3.4-j}}$ (page 19, line 406), $\boxed{\textbf{R1:Ma3.4-k}}$ (page 19, line 410) and $\boxed{\textbf{R1:Ma3.4-l}}$ (page 21, line 445). The authors want to remark, that for the present investigations, which are a basis for the subsequent investigations of the wind turbine including flaps, the focus was not on the exact reproduction of the unsteady inflow conditions. This will be done in future investigations and is now mentioned in the manuscript, see $\boxed{\textbf{R1:Ma3.4-e}}$ (page 18, line 392).

3.5 *"Section 2.1 'Experimental setup' does not describe the velocity measurement setup."*

The velocity measurement setup was already described in sub-subsection 3.1 together with the description of the approach for the FLOWer simulations, see $\boxed{\textbf{R1:Ma3.5}}$ (page 13, line 286). The authors preferred to put the descriptions of the data acquisition for experiment and simulations in one subsection rather than divide them into different subsections.

3.6 *"Mention the specific acquisition devices, and not just the name of the manufacturer."*

Specific acquisition devices can now be found at $\boxed{\textbf{R1:Ma3.6-a}}$ (page 8, line 150), $\boxed{\textbf{R1:Ma3.6-b}}$ (page 8, line 151) and $\boxed{\textbf{R1:Ma3.6-c}}$ (page 8, line 162).

4. *"P5 Figure 3: The actuators for the flaps and the 3-hole probes + air tubes on the smart blade look like they will influence the airflow around the blade. The presence and impact of this blockage should be discussed."*

Information about the influence of the probes, their holder and the tubing was added, see $\boxed{\textbf{R1:Ma4-a}}$ (page 8, line 164). Moreover, a reason for the neglect in the simulation was added, too, see $\boxed{\textbf{R1:Ma4-b}}$ (page 13, line 271).

5. *"P10 L3: An acquisition time of 16 seconds is short for velocity measurements. Please motivate, e.g. based on the integral time scale, that this is sufficiently long for good statistics."*

The time is assumed to be long enough for good statistics for the current setting as the measured integral length scale is $\leq 0.15m$. With the inflow velocity of 6.5m/s as convective velocity, an integral time of t = 0.15m / 6.5m/s = 0.023s is achieved, which is considerably smaller than the acquisition time of 16s. This information was added in the manuscript, see $\boxed{\textbf{R1:Ma5}}$ (page 14, line 289).

6. *"P15 L5: I don't agree with the statement that the error is small. The error in figure 12 is higher than 10% in a large part of the wake: shear region and center."*

Fig. 12 in the first submission corresponds to Fig. 11 in the present submission.
The authors agree and reformulated the sentence see $\boxed{\textbf{R1:Ma6-a}}$ (page 20, line 429). Moreover, the wording "quite good" was replaced with "acceptable", see $\boxed{\textbf{R1:Ma6-b}}$ (page 21, line 436). Additionally, as already mentioned in relation to Major Comment Ma3.4, the mean differences between measurement and simulation were determined and mentioned in the text, see $\boxed{\textbf{R1:Ma3.4-h}}$ (page 18, line 395) and $\boxed{\textbf{R1:Ma3.4-i}}$ (page 20, line 430).

7. "*Figure 18: The experiments show a significant dip around 90 degrees. This is not visible in the simulations. Is there a reason for this effect? Is this also due to the traverse? Explain the situation.*"

This effect is also due to the traverse, which was located upstream of the rotor during all measurements. The influence of the traverse is now mentioned for CaseYAW15, see $\boxed{\textbf{R1:Ma7-a}}$ (page 26, line 537), and CaseYAW30, too, see $\boxed{\textbf{R1:Ma7-b}}$ (page 28, line 554).

8. "*The differences between the measured and simulated bending moments in figures 20 and 21 are significant. It is not ok to say that this is a good agreement. The experimental curves follow a different pattern, especially for CaseBase. Is there an explanation for this?*"

The authors agree, that the differences between the measured and simulated bending moments are significant.
Strong fluctuations are visible in the raw data of the measured bending moments and heavy filtering was necessary to obtain the distributions shown in the first version of this manuscript. The resulting data should only be used for qualitative comparison to numerical results but cannot be considered as valid basis for quantitative comparisons and code validation purposes. We therefore decided, based on Major Comment Ma2 for reviewer 2, to discard all measured bending moments in the revised version of the manuscript.
The reason for the removal is given at $\boxed{\textbf{R1:Ma8-a}}$ (page 16, line 351) and the text was adopted and the corresponding passages were removed, see $\boxed{\textbf{R1:Ma8-b}}$ (page 1, line 12), $\boxed{\textbf{R1:Ma8-c}}$ (page 1, line 13), $\boxed{\textbf{R1:Ma8-d}}$ (page 3, line 80), $\boxed{\textbf{R1:Ma8-e}}$ (page 29, line 569), $\boxed{\textbf{R1:Ma8-f}}$ (page 30, line 582), $\boxed{\textbf{R1:Ma8-g}}$ (page 31, line 590), $\boxed{\textbf{R1:Ma8-h}}$ (page 33, line 639), $\boxed{\textbf{R1:Ma8-i}}$ (page 33, line 644) and $\boxed{\textbf{R1:Ma8-j}}$ (page 33, line 646). The corresponding figures (Fig. 19 and Fig. 20) were adopted, too.
As one of the objectives is the comparison of a medium and a high fidelity code, we consider a comparison of the bending moments calculated with the two numerical methods important.
However, the QBlade results were revised and improved in the course of another paper (Marten et al. 2018). They were replaced in the present manuscript, too, to provide the latest results. In this concerning paper, the present manuscript was cited.
In the former QBlade simulations, the size of the vortex was estimated too large. Instead of the time offset, the parameter "initial vortex core size" is used now in the vortex evolution equation. This parameter is more common in literature and better defined. In the present investigation, approximately 10% midspan chord are used for this parameter, leading to a 50% smaller vortex core. The relevant parameters for the QBlade simulation are now listed in Table 3.
The corresponding Figures (Fig. 19 and Fig.20) were adopted.

9. "*The authors should be careful with copyrights. For instance figure 1a, figure 2, figure 3b and figure 8 can be found identically in the paper 'Reproducible inflow modifications for a wind tunnel mounted research HAWT' by Bartholomay S., et al. 2017.*"

Thanks a lot for this information. The authors are in contact with ASME concerning the copyright. But as the pictures show setups and approaches and no result graphs, it should not cause an infringement. However, Fig. 3 (former Fig.2) was replaced and Fig. 4 (former Fig. 3b) is slightly changed, to have less identical pictures.

If needed, the pictures can also be completely replaced.

10. *"P24 L26: The main conclusion of this paper is too strong. The experiments have too many differences (e.g. vertical shear and turbulence) to make this statement. Furthermore, the agreement for the bending moment is not good at all. Instead make conclusions on what can be estimated correctly, what not, and which physics are modelled correctly."*

The conclusion was completely revised and the authors tried to make the conclusion less strong, see $\boxed{\textbf{R1:Ma10-a}}$ (page 33, line 623), $\boxed{\textbf{R1:Ma10-c}}$ (page 33, line 624), $\boxed{\textbf{R1:Ma10-e}}$ (page 33, line 626) and $\boxed{\textbf{R1:Ma10-n}}$ (page 33, line 648).

Additionally, we extended the conclusions to quantify the differences between simulation and measurement, see $\boxed{\textbf{R1:Ma10-b}}$ (page 33, line 624), $\boxed{\textbf{R1:Ma10-d}}$ (page 33, line 625), $\boxed{\textbf{R1:Ma10-i}}$ (page 33, line 633) and $\boxed{\textbf{R1:Ma10-j}}$ (page 33, line 635) in order to find out, how good the parameters can be estimated.

Moreover, the according reasons for the differences were added, see $\boxed{\textbf{R1:Ma10-g}}$ (page 33, line 626) and $\boxed{\textbf{R1:Ma10-s}}$ (page 34, line 658)

Furthermore, information about the modelled physics are extended and added, see $\boxed{\textbf{R1:Ma10-t}}$ (page 32, line 611), and $\boxed{\textbf{R1:Ma10-u}}$ (page 32, line 613) as well as $\boxed{\textbf{R1:Ma10-h}}$ (page 33, line 641) and $\boxed{\textbf{R1:Ma10-o}}$ (page 33, line 649).

**Minor comments "Mi"**

1. *"Define abbreviations at first use in the main text. Don't define abbreviations in the abstract, and limit the use of abbreviations in the abstract. For example CFD, LLFVW, URANS, .."*
The abbrevations are removed from the abstract and inserted at the first use in the main text, see $\boxed{\textbf{R1:Mi1-a}}$ (page 1, line 7), $\boxed{\textbf{R1:Mi1-b}}$ (page 1, line 8), $\boxed{\textbf{R1:Mi1-c}}$ (page 1, line 10), $\boxed{\textbf{R1:Mi1-e}}$ (page 1, line 11), $\boxed{\textbf{R1:Mi1-f}}$ (page 1, line 11), $\boxed{\textbf{R1:Mi1-g}}$ (page 1, line 13), $\boxed{\textbf{R1:Mi1-h}}$ (page 2, line 37), $\boxed{\textbf{R1:Mi1-i}}$ (page 3, line 72) and $\boxed{\textbf{R1:Mi1-j}}$ (page 2, line 54).

2. *"Throughout the text, several sentences are unnecessarily long, or have a structure where the subject is placed at the end, which can be confusing. Improving these sentences will benefit the clarity of the text."*

The complete manuscript was revised to make the wording more concise.

*"Some examples are:"*
2.1 *"P1, L1"*

The syntax was changed, see $\boxed{\textbf{R1:Mi2.1}}$ (page 1, line 1)

2.2 *"P1, L2: This is a long sentence and not entirely clear. For instance 'methods of different fidelity' is too vague."*

The sentence was cut into several short sentences, see $\boxed{\textbf{R1:Mi2.2-a}}$ (page 1, line 2), $\boxed{\textbf{R1:Mi2.2-b}}$ (page 1, line 2), $\boxed{\textbf{R1:Mi2.2-d}}$ (page 1, line 4) and $\boxed{\textbf{R1:Mi2.2-e}}$ (page 1, line 3). Moreover, 'methods of different fidelity' is more specified, see $\boxed{\textbf{R1:Mi2.2-c}}$ (page 1, line 5)

2.3 *"P1, L5: Is it relevant where the code was run?"*

The information was inserted to make clear, which part of the data was created at which institution and can therefore be assigned to the authors. We therefore prefer to keep this information.

2.4 *"P2, L3: long sentence."*

The sentence was reformulated and split into two sentences, compare R1:Mi2.4-a (page 3, line 68) and R1:Mi2.4-c (page 3, line 69).

3. *"P2 L11: It is not clear what 'a one third model' is + this is a long sentence."*

Information about the one third model is added and the sentence is splitted into two sentences, see R1:Mi3-a (page 2, line 55).

4. *"P2 L17: The yaw angle is negative. Does this matter? The orientation is not mentioned."*

Due to the revision of the introduction, the sentence was omitted.
However, the word "clockwise" was added at the description of the yaw cases, see R1:Mi4 (page 4, line 95). In the experiment, the door to the settling chamber would have been blocked if the turbine would have been rotated counter clockwise.

5. *"P2 L19: 'the flow around the rotor' is too vague."*

Due to the revision of the introduction, the sentence was omitted.

6. *"Define the term 'far field conditions'."*

A short explanation of what is meant by 'far field condition' is added, see R1:Mi6 (page 11, line 239).

7. *"Units need to be formatted correctly."*

The format of the units were changed throughout the whole manuscript. For reasons of simplicity, it is only marked with R1:Mi7-a (page 5, line 117) and R1:Mi7-b (page 13, line 286) exemplary.

8. *"The first sentence of the introduction is too vague and unnecessary."*

The first sentence was deleted and the following sentence was adjusted, see R1:Mi8 (page 1, line 18).

9. *"P1 L17: You mention 'simulations' but the referenced paper presents experimental results."*

That was an infelicitous wording. The mentioned paper was only a reference for the MEXICO project and not for the simulations of the MEXICO rotor in particular. The authors apologize for the confusion. The sentence was adopted and an exemplary reference for the simulation of the MEXICO experiments was added, see R1:Mi9-a (page 2, line 34) and R1:Mi9-b (page 2, line 38).

10. *"P2 L16: Instead of 'three different states', it would be more clear to mention 'Three different yaw-misalignment cases'."*

The sentence was changed, see R1:Mi10 (page 3, line 76).

11. *"P2 L23: This sentence is very long, consider breaking it up in several more clear and well defined sentences."*

The sentence was revised and split up in shorter sentences, see R1:Mi11-a (page 3, line 64), R1:Mi11-c (page 3, line 66) and R1:Ma1-e (page 3, line 65).

12. *"P3 L6: The text mentions 'low Reynolds numbers'. Please describe the Reynolds numbers at which the experiments are run, and motivate if the experiments scale realistically."*

The main goal of the turbine is to deliver data for the comparison to simulations and to test and analyze flow control devices and not to compare the overall performance to a turbine in the free field. Therefore, a low Reynolds airfoil was selected as it provides attached flow in the root region of the turbine, which is better for the comparability of experiment and simulation. Over the whole blade span, the Reynolds number ranges from Re=170000 at 15%R over Re=276000 at 50%R to Re=162000 at 98%R. The Reynolds numbers at 15%R and 75%R are now provided, see $\boxed{\textbf{R1:Mi12-c}}$ (page 6, line 133) and Table 2.

The load alleviation concepts will be investigated in future studies and this submission serves as a basis. In the manuscript, additional information concerning the airfoil see $\boxed{\textbf{R1:Mi12-a}}$ (page 6, line 132), $\boxed{\textbf{R1:Mi12-b}}$ (page 6, line 132) and the motivation of the scaling, see $\boxed{\textbf{R1:Mi12-d}}$ (page 6, line 140), were added.

13. *"P3 L9: How is the boundary layer thickness estimated? Is it possible to indicate the tape on figure 3?"*

That was a mistake the authors want to apologize for. The turbulator heights for the model wind turbine were adopted to the Reynolds number, which changes with blade radius. It was estimated with the help of an additional 2D experiment. Thereby, the Reynolds number over the whole blade radius was determined and reproduced in the Model Wind Tunnel (MWT) of the IAG, University of Stuttgart. With the help of a stethoscope, the state of the flow was investigated behind zig-zag tapes with different heights. Based on these investigations, the turbulator heights for the BeRT turbine were estimated in order to get a defined transition position and to avoid overtripping. The sentence was adopted and a short remark, that the heights were estimated experimentally was added in the submission, see $\boxed{\textbf{R1:Mi13}}$ (page 6, line 136).

The tape position is now indicated in the corresponding figure (Fig. 4, former Fig. 3).

14. *"P3 L15: It is confusing to mention at this point in the text the overall goal of the research project, as it is different from the objectives of this paper."*

The sentences was reformulated and refers now to the aims of the manuscript rather than the aims of the overall project, see $\boxed{\textbf{R1:Mi14}}$ (page 6, line 145).

15. *"P5 L2: What is meant with 'trailing edge deployment'?"*

The smart blade has trailing edge flaps. The actuators are used to deflect the flaps and the sensors to monitor the deflection. So the word 'flap' was missing and is now added, see $\boxed{\textbf{R1:Mi15}}$ (page 8, line 158).

16. *"Table 2: What are the units for the wake length?"*

The wake length is measured in rotor revolutions after release of the respective wake elements. A wake length of two means that a wake element is removed from the domain after the rotor completes two full revolutions after it has been released from the blades trailing edge. Additional information was added in the text, see $\boxed{\textbf{R1:Mi16}}$ (page 10, line 216).

17. *"P6 L14: 21 panels are mentioned in the text, but in Table 2 15 panels are mentioned. Which one is correct?"*

The final calculations have been carried out with 21 blade panels. It was corrected in the table, see Table 3 (former Table 2).

18. "*P7 l3: It is not entirely clear what is meant with 'overlapped using the CHIMERA technique', overlapping several grids?*"

The sentence was extended, see R1:Mi18 (page 11, line 232). And yes, several grids overlapp, as each component of the wind turbine has a separate grid. Afterwards, the single components are put together and the grids are overlapped. The flow of information between the single meshes is handled with the help of the so called Chimera technique. More information about the technique can be found in the corresponding reference. The most important advantages of the technique are the possibility to adopt the mesh for every body to the corresponding requirements of the body and the possibility to move the grids against each other.

19. "*P7 L7: Don't use double brackets '( () )'.*"

The double brackets were removed, see R1:Mi19 (page 11, line 236).

20. "*Table 3: I suppose the units are millions of cells?*"

Yes, that is correct. The information can be found in the table header, but it was now added in first column of Table 4, (former Table 3), too.

21. "*P 8 L6-7: This is a repetition.*"

The sentence was deleted and some of the information are placed in the next sentence, see R1:Mi21 (page 12, line 248).

22. "*Table 4 is an important table. Maybe it can be discussed earlier in the text.*"

Indeed, the table is important and helps to clarify the different cases. Therefore, the whole subsection including Table (Table 1, former Table 4) is now placed at the beginning of the section, see R1:Mi22-a (page 4, line 90) and R1:Mi22-b (page 4, line 103). Moreover, the title of the section was changed so its content is more obvious and information about the position of the nozzle were added.

23. "*P10 L3: Which probe was taken as the reference then? How are the hot-wire probes calibrated?*"

The mean value of all four probes was calculated and used as reference for each measurement position. Additional information about the offset correction and the calibration of the probes were added, see R1:Mi23 (page 14, line 292).

24. "*Figure 7 is unnecessary.*"

The figure was deleted.

25. "*P11 L15: How much are these corrections typically? Maybe indicate in figure 16.*"

The sentence was reformulated in order to make it more understandable. Moreover, an approximated linear equation for the conversion of the local flow angle at the probe to the actual AoA, which is valid in the linear regime, was added, see R1:Mi25 (page 15, line 323) and equation 1.
The authors preferred this way of displaying the conversion instead of the indication in a figure. Regarding Ma2, additional information about error was added in the revised manuscript, see R1:Ma2-a (page 15, line 329).

26. "*P12: The description of the strain gauge setups should be done in the experimental setup section.*"

The strain gauge is now mentioned in the section 2.1.3 , see R1:Mi26 (page 8, line 168), but no more detailed description is present, as the measured bending moments are no longer part of the submission.

27. "*P9 L17 The text mentions measurements at 1.05d , while P13 L3 doesn't mention measurements at 1.05D. Be consistent, also with the unit of 'D'.*"

All positions are now designated with a lower-case "d", according to Fig. 2.
The measurements were performed at three positions. Therefore, for reasons of completeness, the authors mentioned in the manuscript. But because of space reasons, only two locations were analyzed in the submission. Moreover, the additional position at 1.05d would not have brought further benefit for the manuscript.
An additional sentence which explains this fact is added, see $\boxed{\textbf{R1:Mi27}}$ (page 16, line 364).

28. "*P13 L15 'More information about this topic can be found in..' is too vague.*"

It is now more specified what can be found in the provided reference, see $\boxed{\textbf{R1:Mi28}}$ (page 17, line 381).

29. "*P14 L6: 'Some aspects' is too vague.*"

The aspects are now more specified, see $\boxed{\textbf{R1:Mi29}}$ (page 18, line 401).

30. "*P14 L10-15: Conclusions on wake comparison are not clear, which of the two simulations is discussed?*"

A sentence is added in subsection 3.1 to make clear, that only the velocity planes from the simulation including wind tunnel are taken into account in the whole submission, see $\boxed{\textbf{R1:Mi30-a}}$ (page 14, line 302). Moreover, in the text where the comparison is made, it is also mentioned that the comparison to the simulation including wind tunnel is drawn, see $\boxed{\textbf{R1:Mi30-b}}$ (page 19, line 406).

31. "*P15 L12: What does this mean for the measurement blade: isn't 100% the maximal radial position?*"

You are right, 100% is the maximal radial position of the blade, which corresponds to 1.5m and represents the tip. Consequently, 0% corresponds to the center of the rotor. This information is now added, compare $\boxed{\textbf{R1:Mi31-a}}$ (page 21, line 441) and $\boxed{\textbf{R1:Mi31-b}}$ (page 21, line 442).

32. "*P17 L8-10: 'More information about..' is too vague.*"

The intention of the authors to mention this reference was to give the reader a possible reference, where they can find detailed information about all the effects which occur under yawed inflow. The sentence is now changed, see $\boxed{\textbf{R1:Mi32}}$ (page 23, line 488).

33. "*P18 L10: 'More information about..' is too vague.*"

The authors could not explain all the details about the different methods to extract the on-blade velocity and the angle of attack. However we wanted to give information about further literature. The authors now changed the way the reference is mentioned in the text, see $\boxed{\textbf{R1:Mi33}}$ (page 26, line 518).

34. "*P18 L13-L16: This should be mentioned in the introduction.*"

The sentences were move in the introduction, see $\boxed{\textbf{R1:Mi34-a}}$ (page 2, line 47) and $\boxed{\textbf{R1:Mi34-b}}$ (page 2, line 42)

**Reply to the comments of Reviewer No. 2**

Annette Claudia Klein on behalf of the authors
IAG, University of Stuttgart

March 31, 2018

The authors would like to thank the reviewer for his/her efforts and constructive comments. They are very much appreciated and incorporated into the revised submission.

In this document the comments given by the 2nd reviewer are addressed consecutively. The following formatting is chosen:

- The reviewer comments are marked in blue and italic.

- The reply by the authors is in black color.

- A marked-up manuscript is added. Changed sections with regard to the comments by reviewer 2 are marked in orange. Changed sections with regard to comments by both reviewers are marked in gray. Changes with regard to no comments but which serve a better understanding and an improvement of the manuscript are marked in green.

Some manuscripts, which were accepted during the review of this manuscript, are now published:
Fischer et al. 2016 is now referred under Fischer et al. 2018
Wendler et al. (2016) is now published.
Klein et al. (2017) is now referred under Klein et al. (2018)

Moreover, Jost (2017) and Klein (2017) are now replaced by Jost et al. (2018)
The display of the references were adopted in the reference list.

We would like to mention, that since the first submission of this manuscript in September 2017, two conference papers, which partly use the same data as in the present manuscript, were written, submitted and accepted for the AIAA 2018 Conference Series. As these papers reference on the present submission, they were not cited here.

Bartholomay, Sirko, et al. "Towards Active Flow Control on a Research Scale Wind Turbine Using PID controlled Trailing Edge Flaps." 2018 Wind Energy Symposium. 2018.

Marten, David, et al. "Numerical and Experimental Investigation of Trailing Edge Flap Performance on a Model Wind Turbine." 2018 Wind Energy Symposium. 2018.

**General comments "G"**

1. *"Unfortunately, the paper suffers from major issues that require a major revision before the paper can be considered for the publication."*

The authors apologize. The manuscript was completely revised and all comments were considered.

**Major comments "Ma"**

1. "*The introduction is incomplete. The authors only provided a review of the previous studies done by themselves and their colleagues, and they neglected the key papers and contributions done by the other researchers who worked extensively on this topic (e.g., Research group at JHU, EPFL, KU-Leuven, ...).*"

The literature overview was extended and the introduction was completely revised. References about hot-wire measurement to investigate the wake under different operating conditions like yaw misalignment can be found at $\boxed{\textbf{R2:Ma1-a}}$ (page 1, line 22). Further information about the measurement of mean velocity and turbulence intensity was integrated at $\boxed{\textbf{R2:Ma1-b}}$ (page 2, line 25). References about further applications and benefits of wind tunnels can now be found at $\boxed{\textbf{R2:Ma1-c}}$ (page 2, line 31) and references about the investigation of the blockage ratio were now added at $\boxed{\textbf{R2:Ma1-d}}$ (page 2, line 43) and $\boxed{\textbf{R2:Ma1-e}}$ (page 2, line 49).

2. "*The objective of the work is performing numerical simulations to validate the experimental data. However, the results presented in the paper cannot be considered as a validation. There is a huge discrepancy between the experimental data and numerical results and the authors did not explain the reasons behind that. The authors should perform systematic experimental and numerical experiments with providing a clear explanation of the observed discrepancies. Similar studies have been extensively performed by the other groups which some of them mentioned above. For example, the results presented in Figure 21 cannot be considered as validation. There is a huge difference between the experimental data and simulation results. Besides, the authors mentioned: "The curve for the baseline blade is missing in the current plot as the sensors had a malfunction during the measurement." This statement is not acceptable for a paper that is going to be published in a journal. The same trend is also presented in the other figures. No clear explanations are provided about the differences.*"

We agree with the reviewer's assessment with respect to the measurements of the bending moment but are confident with the on-blade velocity, AoA and flow field measurements.
Strong fluctuations are visible in the raw data of the measured bending moments and heavy filtering was necessary to obtain the distributions shown in the first version of this manuscript. We agree that the resulting data should only be used for qualitative comparison to numerical results but cannot be considered as valid basis for quantitative comparisons and code validation purposes. We therefore decided to discard all measured bending moments in the revised version of the manuscript.
The reason for the removal is given at $\boxed{\textbf{R2:Ma2-a}}$ (page 16, line 351). Moreover, the text was adopted and the corresponding passages were removed, see $\boxed{\textbf{R2:Ma2-b}}$ (page 1, line 12), $\boxed{\textbf{R2:Ma2-c}}$ (page 1, line 13), $\boxed{\textbf{R2:Ma2-d}}$ (page 3, line 80), $\boxed{\textbf{R2:Ma2-e}}$ (page 29, line 569) and $\boxed{\textbf{R2:Ma2-f}}$ (page 33, line 639). The corresponding figures (Fig. 19 and Fig. 20) were adopted, too.
As one of the objectives is the comparison of a medium and a high fidelity code, we consider a comparison of the bending moments calculated with the two numerical methods important. Especially for future simulations including flaps, which will be explained in more detail in the next Major Comment (Ma3).
However, the QBlade results were revised and improved in the course of another Paper (Marten et al. 2018). They were replaced in the present manuscript, too, to provide the latest results.

In this concerning paper, the present manuscript was cited. In the former QBlade simulations, the size of the vortex was estimated too large. Instead of the time offset, the parameter "initial vortex core size" is used now in the vortex evolution equation. This parameter is more common in literature and better defined. In the present investigation, approximately 10% midspan chord are used for this parameter, leading to a 50% smaller vortex core. The relevant parameters for the QBlade simulation are now listed in Table 3.

The authors consider the experimental data for the on-blade velocity and the AoA suited for validation, as the progression of the curves and the differences between simulation and experiment can be explained. Explanations which were already included in the first submission of the manuscript can be found, for example, at $\boxed{\textbf{R2:Ma2-h}}$ (page 21, line 450) or $\boxed{\textbf{R2:Ma2-i}}$ (page 26, line 521). The differences between the measured and simulated velocity planes are bigger, but can still be explained, see for example $\boxed{\textbf{R2:Ma7-b}}$ (page 16, line 373) or $\boxed{\textbf{R2:Ma2-j}}$ (page 17, line 380).

More explanations about the differences are now added in the text, see Major Comment Ma4 (e.g. $\boxed{\textbf{R2:Ma4-g}}$ (page 20, line 430)) or Ma5 (e.g. $\boxed{\textbf{R2:Ma5-g}}$ (page 22, line 464)).

References of the groups proposed by the reviewer, as well as further references were added, see major comment 1 (Ma1).

3. "*The authors mentioned: "As it is currently not possible to include the wind tunnel walls into the LLFVW simulations of QBlade, far-field simulations were conducted." Since the objective of the paper is exploring the blockage effect, it is not clear what the purpose of having the results from the QBlade is. It would be more relevant if the QBlade results considering the wind tunnel wall are added to the paper. Otherwise, it is not needed to add the results from another code that is not consistent with the experimental investigation.*"

The exploration of the blockage effect is only one of the submissions's objectives. Actually, the aims of the present study are threefold, compare the revised abstract ($\boxed{\textbf{R2:Ma3-a}}$ (page 1, line 2)) and introduction ($\boxed{\textbf{R2:Ma3-b}}$ (page 3, line 64)). One of these aims is the "comparison and evaluation of methods of high fidelity like Computational Fluid Dynamics and medium fidelity like Lifting Line Free Vortex Wake" ($\boxed{\textbf{R2:Ma3-c}}$ (page 1, line 5)), respectively "the comparison of codes with different grades of fidelity" ($\boxed{\textbf{R2:Ma3-d}}$ (page 3, line 66)). This objective was achieved as it could be shown, that the lack of wind tunnel walls in the QBlade simulation only led to a constant offset regarding on-blade velocity and AoA. This is an important information concerning further investigations with QBlade.

Unfortunately, as mentioned in the manuscript ($\boxed{\textbf{R2:Ma3-e}}$ (page 9, line 196)), the consideration of wind tunnel walls in QBlade is not possible, yet. Therefore, no QBlade results including the wind tunnel walls are available so far.

However, one of the advantages of QBlade is the fact, that it can produce results very fast compared to CFD codes. Therefore, it is well suited, for example, for parametric investigations, controller design or load calculations etc.. The investigations presented in this submission are performed in the course of the DFG PAK 780 project. One of the objectives of the project is the investigation of active trailing edge flaps. These flaps are already integrated in the model wind turbine, although they are fixed in their neutral position for the experiments discussed in this manuscript ($\boxed{\textbf{R2:Ma3-f}}$ (page 8, line 163)). Subsequent measurements on the model wind turbine including flaps shall be used to validate the implementation of the flaps in the CFD and LLFVW code, so they can be used to build the bridge to full size turbines afterwards.

Though, prior to the investigations including flaps, the simpler case without actuated flaps was investigated and used to determine the differences between experiment and simulation. The results of this investigation are presented in the present submission.

With the knowledge gained by the comparisons, the LLFVW code could be used to determine the ideal flap deflection to mitigate known disturbances, for example caused by yaw misalignment, see the two references at the beginning of this document (Marten et al. 2018 and Bartholomay et al. 2018). These mutual comparisons are only possible with the knowledge about the influence of the wind tunnel walls on the QBlade results, see $\boxed{\textbf{R2:Ma3-g}}$ (page 33, line 618)

Moreover, after the implementation of the flaps in the CFD and LLFVW code has been validated, the codes can be used to investigate full size turbines, where no wind tunnel walls influence the results. In addition to that, with the possibility of QBlade to produce fast results, the code can be used on the one hand for parametric studies to determine the ideal size and position of the flaps as well as the ideal deflection for a previously known disturbance such as yaw misalignment.

To sum up, the results of the comparison of QBlade results to results of a CFD code and to experiments are an important basis for the assessment of the LLFVW code for further investigations of the model wind turbine including flaps.

For these reasons, the authors considered the inclusion of QBlade results relevant as basis for future studies. The authors hope, that these explanations could show the reviewer the importance of the comparisons with QBlade, even as the simulations neglect the effect of the wind tunnel walls.

4. "*As it is mentioned before, the validation section is not acceptable. Also, since Figures 9 and 10 are qualitative results, the authors need to provide more quantitative comparisons by, for instance, comparing the results at different locations. Although, even from the contour plots, the agreement is not good. Also, besides the mean velocity, the variances obtained from both the experiment and numerics should be provided. This is a very standard way for validation of numerical tools against the experimental data.*"

The authors want to apologize for the insufficient validation section. The complete chapter was revised and more qualitative comparisons along with interpretations were added

In the manuscript, the measured and the simulated flow fields are compared at two locations (one upstream and one downstream of the rotor). At another location (+1.05d downstream) further measurements were performed and simulation data for that plane is also available. However, the authors decided to forego the comparison at this location as the evaluation would not have brought further benefit or pursuing information for the submission.

Moreover, at this location, the influence of the nozzle is already present, which influences the wake development further. This is now also mentioned in the text, see $\boxed{\textbf{R2:Ma4-a}}$ (page 16, line 364). But the data are available and could be included.

According to reviewer 1 Major Comment Ma3.4 and $\boxed{\textbf{R2:Ma7-b}}$ (page 16, line 373), more information about the inequalities in the measured inflow velocity are addressed now. Moreover, according to $\boxed{\textbf{R2:Ma7-a}}$ (page 5, line 118), a comment on the turbulent inflow field is added. To consider the reviewer's request for more quantitative results, the authors included tables with the streamwise mean velocity, the standard deviation of the streamwise velocity as well as the global turbulence intensity in x-y-direction for both locations, see Table 5 $\boxed{\textbf{R2:Ma4-b}}$ (page 17, line 383) and Table 6 $\boxed{\textbf{R2:Ma4-e}}$ (page 19, line 418). Compare also reviewer 1 Major Comment Ma 3.4.

We prefer to show the standard deviation instead of the variance to be consistent within the whole manuscript, as the standard deviations are now already inserted for different parameters, like the on-blade velocity, see $\boxed{\textbf{R2:Ma4-s}}$ (page 21, line 451), $\boxed{\textbf{R2:Ma4-t}}$ (page 22, line 476) and $\boxed{\textbf{R2:Ma4-u}}$ (page 24, line 493) and the AoA, see $\boxed{\textbf{R2:Ma4-h}}$ (page 25, line 512), $\boxed{\textbf{R2:Ma4-v}}$ (page 26, line 542) and $\boxed{\textbf{R2:Ma4-w}}$ (page 29, line 562). Moreover, the mean differences between measurement and simulation were determined and mentioned in the text, see $\boxed{\textbf{R2:Ma4-d}}$ (page 18, line 395) and $\boxed{\textbf{R2:Ma4-g}}$ (page 20, line 430).

Moreover, information about the measurement uncertainty is now included in the revised manuscript, see R2:Ma4-cc (page 15, line 329) and R2:Ma4-dd (page 14, line 297). The authors want to remark, that for the present investigations, which are a basis for the subsequent investigations of the wind turbine including flaps, the focus was not on the detailed reproduction of the unsteady inflow conditions. This will be the content of a future study, were the turbulent parameters from the measurement are used to create unsteady inflow conditions for the CFD simulations. This was also mentioned in the submission now, see R2:Ma7-f (page 18, line 392).

Now, quantitative results of the velocity planes are discussed in the text, see R2:Ma4-c (page 17, line 385), R2:Ma4-x (page 17, line 387), R2:Ma4-f (page 19, line 420), R2:Ma4-z (page 19, line 423), as well as R2:Ma4-p (page 33, line 624) and R2:Ma4-q (page 33, line 625). Moreover, quantitative comparisons are added for the other parameters. Concerning the on-blade velocity, see Table 7, 8 and 9, concerning the AoA, Table 10, 11 and 12. The references concerning the discussion of this tables can be found under Major Comment Ma5. Additionally, the statement concerning the agreement was revised, see R2:Ma4-bb (page 21, line 436).

Quantitative comparisons concerning the bending moments as well as explanations can be found at 13 R2:Ma4-i (page 29, line 574) and 14 R2:Ma4-n (page 32, line 592) as well as on R2:Ma4-k (page 29, line 577), R2:Ma4-l (page 30, line 579), R2:Ma4-m (page 30, line 583), R2:Ma4-o (page 32, line 594), R2:Ma4-j (page 29, line 575),

5. "*Figure 13-16, it seems that the y-axis chosen here is too wide to minimize the difference between the experiments and numerics. For example, in Fig. 13 (left), it is trivial that the on-blade velocity cannot be ranged from 0 to 30. In particular, considering the tip-speed ratio and the incoming wind speed, it should be in a much narrower range.*"

The authors assume, that the reviewer refers to Fig. 13-15 (now, due to the suggestions of reviewer 1, Mi24 Fig. 12, Fig. 13 and Fig. 14), as Fig. 16 (now Fig. 15) already shows the angle of attack.

The authors totally agree with the reviewer, that the on-blade velocity cannot range from $0 m/s$ to $30 m/s$. The range of the y-axis in the manuscript reached from $15 m/s$ and $30 m/s$ and the range of the x-axis from $0°$ to $360°$. Due to the proximity of the labels for x- and y-axis, they can easily be mixed-up at the coordinate origin.

Initially, the range of the y-axis from $15 m/s$ to $30 m/s$ was chosen primary for reasons of comparability. Now, the y-axis of Fig. 12, Fig. 13 were adopted to be better adjusted to the occurring velocities. However, the axis range at both radial positions were kept the same for all the cases. Thereby, the level difference between $65\%R$ and $85\%R$ becomes more obvious for each case. Therefore, the axis range of Fig. 14 could not be reduced.

For an even better comparison, the authors added tables (Table 7 R2:Ma5-a (page 22, line 453), Table 8 R2:Ma5-h (page 22, line 477)) and Table 9 R2:Ma5-l (page 24, line 494)) where the average of the differences between experiment and the different simulations and evaluation methods are listed. The undisturbed velocity was chosen as reference, so the cases can be compared over the different yaw angles, as all differences have the same reference velocity, see R2:Ma5-b (page 22, line 456) and R2:Ma5-c (page 22, line 457). Moreover, the contents of the tables are discussed in the text, see R2:Ma5-d (page 22, line 458), R2:Ma5-e (page 22, line 460), R2:Ma5-f (page 22, line 462), R2:Ma5-g (page 22, line 464), R2:Ma5-i (page 23, line 480), R2:Ma5-j (page 23, line 481), R2:Ma5-k (page 23, line 482), R2:Ma5-m (page 24, line 497).

Through this approach, the comparability of the single figures at the different rotor locations remains, but the assessment of the differences of the single curves is improved.

In order to improve the analysis of the angle of attack, too, tables to assess the quantitative differences between experiment and simulation were added as well at Chapter 4.3 (Table 10 R2:Ma5-n (page 25, line 509), Table 11 R2:Ma5-t (page 26, line 539) and Table 12 R2:Ma5-w (page 28, line 556)). Again, the contents of the tables are discussed in the text, see R2:Ma5-o (page 25, line 516), R2:Ma5-p (page 26, line 520), R2:Ma5-q (page 26, line 523), R2:Ma5-r (page 26, line 531), R2:Ma5-s (page 26, line 531), R2:Ma5-u (page 26, line 541), R2:Ma5-v (page 27, line 546), R2:Ma5-x (page 28, line 560), R2:Ma5-y (page 29, line 562) and R2:Ma5-z (page 29, line 564) and additional explanations for the differences are provided, see R2:Ma5-aa (page 26, line 523) and R2:Ma5-bb (page 27, line 547).

Concerning quantitative comparisons of the bending moments, see the references under Major Comment Ma2.

6. "*Most of the citation about the numerical frameworks are technical report, conference proceeding or personal communications. Typically, it is expected that the papers cited in the manuscript were peer-reviewed before.*"

The authors apologize for the lack of peer-reviewed manuscripts on preparatory work for the present study. Journal publications are underway. As there are lots of investigations and further developments are going on right now and the publication in journals take some time, the authors used the technical reports, and conference proceedings to offer the reader further information, whose detailed provision would have been beyond the scope of the present submission.

The authors are happy to inform the reviewer, that one manuscript was accepted in Wind Energy in the meanwhile and will replace two personal communications (Jost, 2017 and Klein, 2017 $->$ Jost et al, 2018) in the present manuscript, see R2:Ma6-a (page 15, line 339). The corresponding text in the manuscript was adopted according to this fact, see R2:Ma6-b (page 15, line 339), R2:Ma6-c (page 15, line 343) and R2:Ma6-d (page 16, line 346).

7. "*The incoming flow is not characterized in the manuscript. The information about the incoming wind, the associated turbulence level, the Reynolds number based on the chord length is missing in the manuscript.*"

Additional information about the inflow is now added in Subsection 2.2.1 see R2:Ma7-a (page 5, line 118) and in Subsection 4.1, see R2:Ma7-b (page 16, line 373). Moreover, related to Major Comment Ma4, further information about the velocity fields were added in the text. A threshold for the turbulence intensity in the settling chamber is provided in the text, see R2:Ma7-c (page 5, line 117) as well as the mean Ti in the velocity planes (compare Table 5 and Table 6).

As suggested by reviewer 1, the Reynolds number at the blade root was added, see R2:Ma7-d (page 6, line 133). Moreover, a representative Reynolds number at $75\%R$ was added in Table 2. This position is located in the middle between the two 3-hole probes, which are used for the determination of the on-blade velocity and the angle of attack.

In future studies, the measured unsteady inflow will be used to create unsteady inflow conditions for the *CFD* simulation. Those planned simulations are now mentioned in the text, see R2:Ma7-f (page 18, line 392).

A picture of the measured horizontal velocity is added in this document.

The horizontal velocity shows the upstream effect of the turbine, as the flow bends to the negative direction for y<0 and in positive direction for y>0. As the horizontal velocity is more than one order of magnitude smaller than the axial velocity, it has only a minor effect on the

[Figure]

Figure 1: Hot-wire measurements of the y-velocity $0.43d$ upstream of the rotor plane. The dashed lines illustrate the wind tunnel and the turbine. Isolines show a velocity of $0.0ms^{-1}$. The dots show the discrete measuring points.

AoA and on-blade velocity. Consequently, it was not shown in the revised submission in order to keep it concise.

8. "*The results are provided without any sensitivity analysis to the grid resolutions. As mentioned before, the agreement between the numerical results and experiment is poor. Although the code might have tested before for other cases, it is required to perform the grid resolution sensitivity for this particular analysis presented in the manuscript. The convergence of the statistics also should be provided*"

The authors agree, that a sensitivity analysis of the grid resolution is very important for CFD simulations. For this reason, a grid convergence study according to Celik et al. (2008), where the dependency of the numerical solution on the grid resolution was estimated, was performed prior to the present investigation. The results were already published in Fischer et al. (2018) and the reference is mentioned in the present manuscript, see $\boxed{\textbf{R2:Ma8-a}}$ (page 12, line 267). The investigation in Fischer et al. (2018) was performed for a one third model of the model wind turbine in a far field environment under uniform inflow. But as the number of cells for the present setup was kept constant or was even increased, for example for the blades, a renewed grid convergence study for the full model was not performed. In order to achieve an easier and better assessment of the sensitivity analysis of the grid resolution for the reader, the authors added further information about the sensitivity analysis from the corresponding paper (Fischer et al., 2018) , see $\boxed{\textbf{R2:Ma8-b}}$ (page 12, line 268).

Overall, 45 rotor revolutions were calculated. The results presented in this manuscript are the averaged data over the last five revolutions. The difference between e.g. the global loads of the rotor averaged over revolution 36-40 to the loads averaged over revolution 41-45 amounts 0.03%. This information was partly already included in the first submission. The missing information was now added and can be found at $\boxed{\textbf{R2:Ma8-c}}$ (page 13, line 276).

**List of the major changes in the manuscript**

The line numbers correspond to the marked-up manuscript, not ot the revised version of the manuscript.

- Abstract
  - page 1, line 1-16: completely revised

- 1 Introduction
  - page 1-3, line 18-78: completely revised, literature overview extended

- 2 Methodology and setup
  - page 3, line 86-88: overview of section added
  - page 4, line 89: section 2.1 pulled forward and change of title
  - page 5, line 106-108: overview of section added
  - page 5-6, line 111-127: information about wind tunnel added
  - page 6, Figure 2: caption extended
  - page 6-8, line 132-151: additional information concerning Reynolds number, main goal of the turbine and blockage ratio added
  - page 7, Figure 3: replaced
  - page 8, line 156: titel of subsection changed
  - page 8, line 162-173: section extended
  - page 9, Figure 4, left: position of zz tape added
  - page 9-10, line 200-212: advantage of LLFVW over BEM added
  - page 10, Table 3: extended
  - page 12-13, line 267-271: information about grid convergence added
  - page 13, line 276-277: number of rotor revolutions added

- 3 Data acquisition
  - page 13, line 279: new subsection
  - page 13, line 280-283: overview of section added
  - page 14, line 289-291: information about measurement time added
  - page 14, line 292-296: information about probe calibration added
  - page 14, line 297-301: information about measurement uncertainty added
  - page 15, line 323-324: information about typical corrections added
  - page 15, equation 1 added
  - page 15, line 329-333: information about measurement uncertainty added
  - page 16, line 351-353: information about lack of measured bending moments added

- 4 Results and discussion
  - page 16, line 364-366: information about additional plane added
  - page 16-17, line 374-377: information about velocity plane added
  - page 17, line 380-382: information about inequalities added
  - page 17-18, line 383-392: information about inflow added
  - page 17, Table 5 added
  - page 19, line 410-413: information about influence of turbulence added
  - page 19-20, line 418-427: quantitative comparison extended

- page 20, Table 6 added
- page 20-21, line 429-435: quantitative comparison extended
- page 21, line 444-450: information about inequalities added
- page 21-22, line 450-462: quantitative comparison extended
- page 22, equation 2 added
- page 22, Table 7 added
- page 22, line 464-473: explanations extended
- page 22-23, line 476-485: quantitative comparison extended
- page 23, Table 8 added
- page 24, line 493-495: quantitative comparison extended
- page 24, Table 9 added
- page 24-25, line 498-500: quantitative comparison extended
- page 25, line 509-517: quantitative comparison extended
- page 26, Table 10 added
- page 26, line 521-527: explanations extended
- page 26, line 531-533: quantitative comparison extended
- page 26, line 542-544: quantitative comparison extended
- page 27-28, line 547-552: explanations extended
- page 28, Table 11 added
- page 28, Figure 18, right: name corrected
- page 28-29, line 560-564: quantitative comparison extended
- page 29, Table 12 added
- page 29, line 569-571: overview of section added
- page 29, Table 13 added
- page 30, Figure 19: measured bending moments removed
- page 30-31, line 583-588: quantitative comparison extended
- page 31, Figure 20: measured bending moments removed
- page 32, Table 14 added
- page 32, line 592-603: quantitative comparison extended
- page 32, line 604-607: short summary added

- 5 Summary
  - page 32-34, line 609-659: completely revised

- Achnowledgement
  - page 34, line 670-671: extended

- References
  - page 35-37, line 673-773: several references updated, e.g. Jost et al 2018
  - page 35-37, line 673-773: several new references, e.g. Bastankhah and Porté-Agel 2015

[revised manuscript text omitted]

**2.2.1 Wind tunnel**

The experiments are carried out in the large wind tunnel (*GroWiKa*) of the TU Berlin, Fig. 2 (Bartholomay et al., 2017), R1:Ma3.2-a which is a circuit wind tunnel R1:Ma3.3-a and is driven by a 450kW fan. R1:Ma3.4-a The $2 \times 1.4\text{m}^2$ cross section of the real test section is too small for the model wind turbine, which has a large diameter to realize the investigation of spanwise locally distributed devices for passive and active flow control in future investigations. Therefore, the real test section was shortened and the $4.2 \times 4.2\text{m}^2$ settling chamber of the wind tunnel was extended to a total length of 5m and was then used as measuring section for the model wind turbine. R1:Ma3.2-b This configuration leads to the unusual fact that the nozzle is positioned downstream of the measuring section. R1:Ma3.3-b The velocity in the settling chamber used for the present investigations amounts R1:Mi7-a $6.5\text{ms}^{-1}$ and the R2:Ma7-c turbulence intensity is in average $Ti \leq 1.5\%$ R1:Ma3.4-b R2:Ma7-a 
[revised manuscript text omitted]

---

## Author Response (AR2)

**This document includes:**

1. Point-to-point response to the first reviewer

2. List of the major changes in the manuscript

3. Marked-up manuscript: changed sections with regard to the comments by reviewer 1 are marked in yellow

**Reply to the comments of Reviewer No. 1**

Annette Claudia Klein on behalf of the authors
IAG, University of Stuttgart

May 28, 2018

The authors would like to thank the reviewer for his/her efforts and constructive comments again. They are very much appreciated and incorporated into the revised manuscript.

In this document the comments given by the 1st reviewer are addressed consecutively. The following formatting is chosen:

- The reviewer comments are marked in blue and italic.

- The reply by the authors is in black color.

- A marked-up manuscript is added. Changed sections with regard to the comments by reviewer 1 are marked in yellow.

**Minor comments "Mi"**

1. "*The term 'far field' is mentioned first in the introduction, it should be explained at first use, instead of later in the text.*"

A short explanation has been added where the term 'far field' is first mentioned in the text, see R1:Mi1 (page 3, line 70).

2. "*3.1: The authors mention the integral length scale. How was it measured? Was it measured by integrating the autocorrelation of a hot-wire time signal and by applying Taylor's hypothesis? In that case it would make more sense to directly mention the integral time scale, instead of applying Taylor's hypothesis to find the length scale from the time scale and then inverse applying Taylor's hypothesis in the text to go back to the time scale.*"

The authors agree with the Reviewer! That was a cumbersome description.
The approach was exactly as described by the Reviewer. The sentence was reformulated and adapted to the chronological order of the results, see R1:Mi2 (page 13, line 278). However, both information (time scale and length scale) are still present in the text because thereby, more information can be provided to the reader and no calculation is necessary on their part.

3. "*Throughout the text, the authors should round numbers (e.g. estimates for uncertainty) to only the significant digits. For example, an uncertainty of 1% instead of 1.12%.*"

The percentual data has been changed, see R1:Mi3-a (page 17, line 374), R1:Mi3-b (page 17, line 382), R1:Mi3-c (page 19, line 415), R1:Mi3-d (page 19, line 418), R1:Mi3-e (page 19, line 419), R1:Mi3-f (page 21, line 445), R1:Mi3-g (page 22, line 464), R1:Mi3-h (page 29, line 557), R1:Mi3-i (page 29, line 559), R1:Mi3-j (page 30, line 566), R1:Mi3-k (page 30, line 566), R1:Mi3-l (page 30, line 577), R1:Mi3-m (page 31, line 584), R1:Mi3-n (page 31, line 584), R1:Mi3-o (page 31, line 585), R1:Mi3-p (page 32, line 607) and R1:Mi3-q (page 32, line 608).

4. "*P13L21: typo 'walls was take into account'*"

The typo has been corrected, see R1:Mi4 (page 13, line 290).

5. "*P14L22: Can the authors add a reference or short description to the mention of a 'Linear regime'?*"

The sentence was slightly changed and a short description of the linear regime of the lift polar was added , see R1:Mi5 (page 14, line 311).

6. "*Equation (1): add units, degrees or radians?*"

The units were added, see R1:Mi6 (page 14, line 314).

7. "*P17L5 typo: 'wirer'*"

The typo has been corrected, see R1:Mi7 (page 17, line 369).

8. "*'However, the comparisons between measurement and calculation will be done anyway': this sentence can be removed.*"

The sentence has been removed, see R1:Mi8 (page 17, line 378).

9. "*P25L14-15: If the absolute value of the incoming velocity is very similar with and without blockage, do the authors have any ideas/suggestion what is causing the larger difference for the angle of attack from blockage?*"

The undisturbed inflow velocity is the same. However, the wake downstream of the turbine for the cases with wind tunnel can not expand as under far field condition, as the walls impede the expansion. Consequentely, the velocity in the rotor plane is higher for the case including wind tunnel. This leads to a higher AoA. More information about this topic can be found in Fischer et al. (2018) and in Klein et al. (2018). An additional sentence, see R1:Mi9-a (page 25, line 502), as well as the two references, see R1:Mi9-b (page 25, line 505) were added in the text.

10. "*P30L14-15 This sentence isn't entirely clear to the reviewer.*"

The sentence was reformulated and split into two sentences, see R1:Mi10 (page 30, line 578).

**List of the major changes in the manuscript**

The line numbers correspond to the marked-up manuscript, not ot the revised version of the manuscript.

- 1 Introduction
  - page 1-3, line 70: information about far field added

- 3 Data acquisition
  - page 13, line 278-280: sentences about integral time scale revised
  - page 13, line 290: typo corrected
  - page 14, line 311-313: information about the linear regime on the lift polar added
  - page 14, line 315: units added

- 4 Results and discussion
  - page 17, line 369: typo corrected
  - page 17, line 374: number rounded
  - page 17, line 378: sentence removed
  - page 17, line 382: number rounded
  - page 19, line 415: number rounded
  - page 19, line 418: number rounded
  - page 19, line 419: number rounded
  - page 21, line 445-446: numbers rounded
  - page 22, line 464: number rounded
  - page 25, line 502-503: information about velocity added
  - page 25, line 505-506: references added
  - page 29, line 557: number rounded
  - page 29, line 559: number rounded
  - page 30, line 566: numbers rounded
  - page 30, line 577: number rounded
  - page 30-31, line 579-580: sentence reformulated
  - page 31, line 584: number rounded
  - page 31, line 585: number rounded
  - page 32, line 607: number rounded
  - page 32, line 608: number rounded

[revised manuscript text omitted]